# A GPU-based computational framework that bridges neuron simulation and artificial intelligence

Yichen Zhang [1,8], Gan He [1,8], Lei Ma [1,2,8], Xiaofei Liu[1,3], J. J. Johannes Hjorth[4], Alexander Kozlov[4,5], Yutao He[1], Shenjian Zhang[1], Jeanette Hellgren Kotaleski [4,5], Yonghong Tian [1,6], Sten Grillner [5], Kai Du [7] ✉ & Tiejun Huang [1,2,7]

Biophysically detailed multi-compartment models are powerful tools to explore computational principles of the brain and also serve as a theoretical framework to generate algorithms for artificial intelligence (AI) systems. However, the expensive computational cost severely limits the applications in both the neuroscience and AI fields. The major bottleneck during simulating detailed compartment models is the ability of a simulator to solve large systems of linear equations. Here, we present a novel **D**endritic **H**ierarchical **S**cheduling (DHS) method to markedly accelerate such a process. We theoretically prove that the DHS implementation is computationally optimal and accurate. This GPU-based method performs with 2-3 orders of magnitude higher speed than that of the classic serial Hines method in the conventional CPU platform. We build a DeepDendrite framework, which integrates the DHS method and the GPU computing engine of the NEURON simulator and demonstrate applications of DeepDendrite in neuroscience tasks. We investigate how spatial patterns of spine inputs affect neuronal excitability in a detailed human pyramidal neuron model with 25,000 spines. Furthermore, we provide a brief discussion on the potential of DeepDendrite for AI, specifically highlighting its ability to enable the efficient training of biophysically detailed models in typical image classification tasks.

Deciphering the coding and computational principles of neurons is essential to neuroscience. Mammalian brains are composed of more than thousands of different types of neurons with unique morphological and biophysical properties. Even though it is no longer conceptually true, the "point-neuron" doctrine[1], in which neurons were regarded as simple summing units, is still widely applied in neural computation, especially in neural network analysis. In recent years, modern artificial intelligence (AI) has utilized this principle and developed powerful tools, such as artificial neural networks (ANN)[2]. However, in addition to comprehensive computations at the single neuron level, subcellular compartments, such as neuronal dendrites, can also carry out nonlinear operations as independent computational

[1]National Key Laboratory for Multimedia Information Processing, School of Computer Science, Peking University, Beijing 100871, China. [2]Beijing Academy of Artificial Intelligence (BAAI), Beijing 100084, China. [3]School of Information Science and Engineering, Yunnan University, Kunming 650500, China. [4]Science for Life Laboratory, School of Electrical Engineering and Computer Science, Royal Institute of Technology KTH, Stockholm SE-10044, Sweden. [5]Department of Neuroscience, Karolinska Institute, Stockholm SE-17165, Sweden. [6]School of Electrical and Computer Engineering, Shenzhen Graduate School, Peking University, Shenzhen 518055, China. [7]Institute for Artificial Intelligence, Peking University, Beijing 100871, China. [8]These authors contributed equally: Yichen Zhang, Gan He, and Lei Ma. ✉e-mail: kai.du@pku.edu.cn

units[3–7]. Furthermore, dendritic spines, small protrusions that densely cover dendrites in spiny neurons, can compartmentalize synaptic signals, allowing them to be separated from their parent dendrites ex vivo and in vivo[8–11].

Simulations using biologically detailed neurons provide a theoretical framework for linking biological details to computational principles. The core of the biophysically detailed multi-compartment model framework[12,13] allows us to model neurons with realistic dendritic morphologies, intrinsic ionic conductance, and extrinsic synaptic inputs. The backbone of the detailed multi-compartment model, i.e., dendrites, is built upon the classical Cable theory[12], which models the biophysical membrane properties of dendrites as passive cables, providing a mathematical description of how electronic signals invade and propagate throughout complex neuronal processes. By incorporating Cable theory with active biophysical mechanisms such as ion channels, excitatory and inhibitory synaptic currents, etc., a detailed multi-compartment model can achieve cellular and subcellular neuronal computations beyond experimental limitations[4,7].

In addition to its profound impact on neuroscience, biologically detailed neuron models recently were utilized to bridge the gap between neuronal structural and biophysical details and AI. The prevailing technique in the modern AI field is ANNs consisting of point neurons, an analog to biological neural networks. Although ANNs with "backpropagation-of-error" (backprop) algorithm achieve remarkable performance in specialized applications, even beating top human professional players in the games of Go and chess[14,15], the human brain still outperforms ANNs in domains that involve more dynamic and noisy environments[16,17]. Recent theoretical studies suggest that dendritic integration is crucial in generating efficient learning algorithms that potentially exceed backprop in parallel information processing[18–20]. Furthermore, a single detailed multi-compartment model can learn network-level nonlinear computations for point neurons by adjusting only the synaptic strength[21,22], demonstrating the full potential of the detailed models in building more powerful brain-like AI systems. Therefore, it is of high priority to expand paradigms in brain-like AI from single detailed neuron models to large-scale biologically detailed networks.

One long-standing challenge of the detailed simulation approach lies in its exceedingly high computational cost, which has severely limited its application to neuroscience and AI. The major bottleneck of the simulation is to solve linear equations based on the foundational theories of detailed modeling[12,23,24]. To improve efficiency, the classic Hines method reduces the time complexity for solving equations from $O(n^3)$ to $O(n)$, which has been widely applied as the core algorithm in popular simulators such as NEURON[25] and GENESIS[26]. However, this method uses a serial approach to process each compartment sequentially. When a simulation involves multiple biophysically detailed dendrites with dendritic spines, the linear equation matrix ("Hines Matrix") scales accordingly with an increasing number of dendrites or spines (Fig. 1e), making Hines method no longer practical, since it poses a very heavy burden on the entire simulation.

During past decades, tremendous progress has been achieved to speed up the Hines method by using parallel methods at the cellular level, which enables to parallelize the computation of different parts in each cell[27–32]. However, current cellular-level parallel methods often lack an efficient parallelization strategy or lack sufficient numerical accuracy as compared to the original Hines method.

Here, we develop a fully automatic, numerically accurate, and optimized simulation tool that can significantly accelerate computation efficiency and reduce computational cost. In addition, this simulation tool can be seamlessly adopted for establishing and testing neural networks with biological details for machine learning and AI applications. Critically, we formulate the parallel computation of the Hines method as a mathematical scheduling problem and generate a Dendritic Hierarchical Scheduling (DHS) method based on combinatorial optimization[33] and parallel computing theory[34]. We demonstrate that our algorithm provides optimal scheduling without any loss of precision. Furthermore, we have optimized DHS for the currently most advanced GPU chip by leveraging the GPU memory hierarchy and memory accessing mechanisms. Together, DHS can speed up computation 60-1,500 times (Supplementary Table 1) compared to the classic simulator NEURON[25] while maintaining identical accuracy.

To enable detailed dendritic simulations for use in AI, we next establish the DeepDendrite framework by integrating the DHS-embedded CoreNEURON (an optimized compute engine for NEURON) platform[35] as the simulation engine and two auxiliary modules (I/O module and learning module) supporting dendritic learning algorithms during simulations. DeepDendrite runs on the GPU hardware platform, supporting both regular simulation tasks in neuroscience and learning tasks in AI.

Last but not least, we also present several applications using DeepDendrite, targeting a few critical challenges in neuroscience and AI: (1) We demonstrate how spatial patterns of dendritic spine inputs affect neuronal activities with neurons containing spines throughout the dendritic trees (full-spine models). DeepDendrite enables us to explore neuronal computation in a simulated human pyramidal neuron model with ~25,000 dendritic spines. (2) In the discussion we also consider the potential of DeepDendrite in the context of AI, specifically, in creating ANNs with morphologically detailed human pyramidal neurons. Our findings suggest that DeepDendrite has the potential to drastically reduce the training duration, thus making detailed network models more feasible for data-driven tasks.

All source code for DeepDendrite, the full-spine models and the detailed dendritic network model are publicly available online (see Code Availability). Our open-source learning framework can be readily integrated with other dendritic learning rules, such as learning rules for nonlinear (full-active) dendrites[21], burst-dependent synaptic plasticity[20], and learning with spike prediction[36]. Overall, our study provides a complete set of tools that have the potential to change the current computational neuroscience community ecosystem. By leveraging the power of GPU computing, we envision that these tools will facilitate system-level explorations of computational principles of the brain's fine structures, as well as promote the interaction between neuroscience and modern AI.

## Results
### Dendritic Hierarchical Scheduling (DHS) method
Computing ionic currents and solving linear equations are two critical phases when simulating biophysically detailed neurons, which are time-consuming and pose severe computational burdens. Fortunately, computing ionic currents of each compartment is a fully independent process so that it can be naturally parallelized on devices with massive parallel-computing units like GPUs[37]. As a consequence, solving linear equations becomes the remaining bottleneck for the parallelization process (Fig. 1a–f).

To tackle this bottleneck, cellular-level parallel methods have been developed, which accelerate single-cell computation by "splitting" a single cell into several compartments that can be computed in parallel[27,28,38]. However, such methods rely heavily on prior knowledge to generate practical strategies on how to split a single neuron into compartments (Fig. 1g–i; Supplementary Fig. 1). Hence, it becomes less efficient for neurons with asymmetrical morphologies, e.g., pyramidal neurons and Purkinje neurons.

We aim to develop a more efficient and precise parallel method for the simulation of biologically detailed neural networks. First, we establish the criteria for the accuracy of a cellular-level parallel method. Based on the theories in parallel computing[34], we propose three conditions to make sure a parallel method will yield identical solutions as the serial computing Hines method according to the data

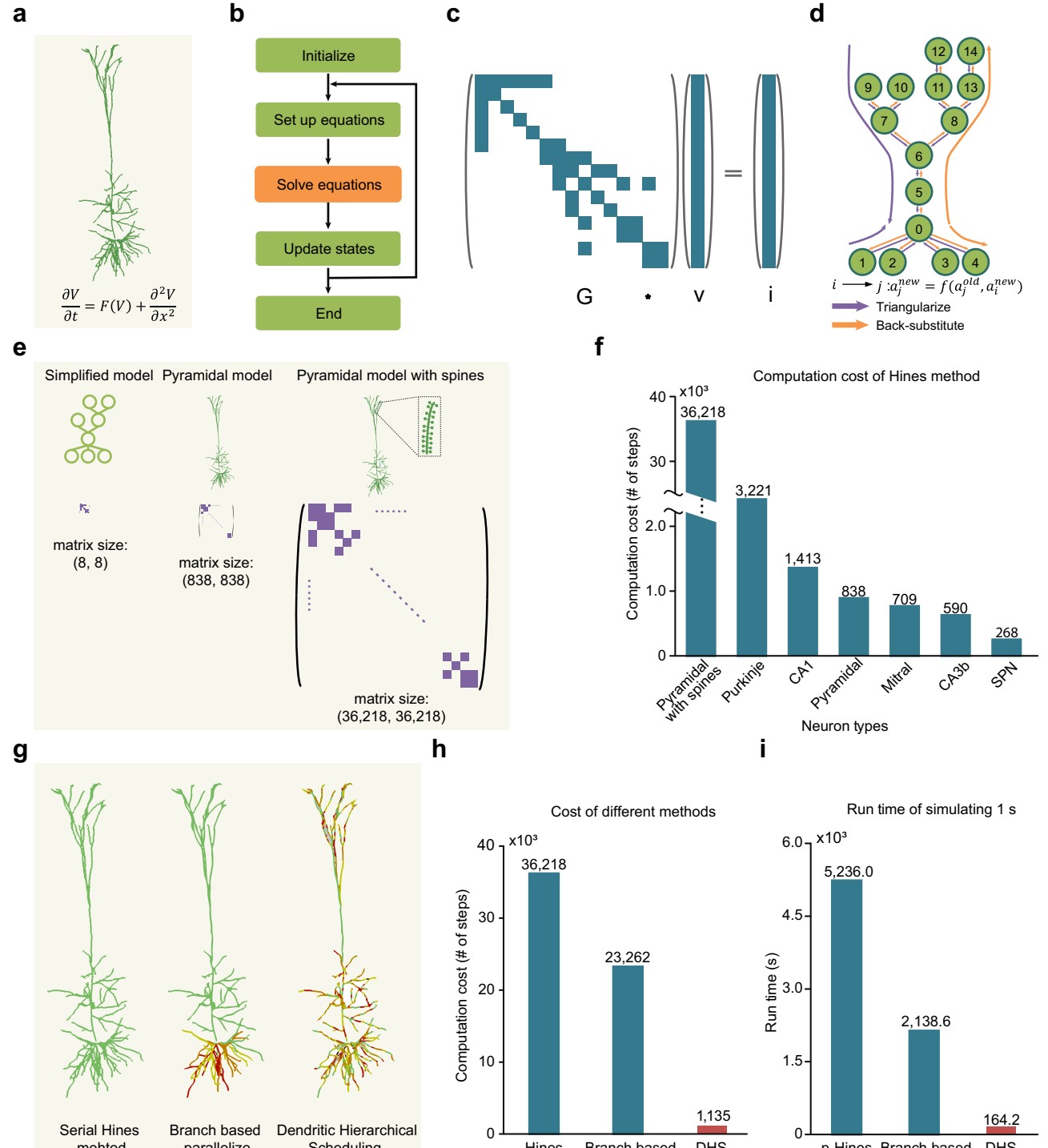

**Fig. 1 | Accelerate simulation of biophysically detailed neuron models. a** A reconstructed layer-5 pyramidal neuron model and the mathematical formula used with detailed neuron models. **b** Workflow when numerically simulating detailed neuron models. The equation-solving phase is the bottleneck in the simulation. **c** An example of linear equations in the simulation. **d** Data dependency of the Hines method when solving linear equations in **c**. **e** The size of the Hines matrix scales with model complexity. The number of linear equations system to be solved undergoes a significant increase when models are growing more detailed. **f** Computational cost (steps taken in the equation solving phase) of the serial Hines method on different types of neuron models. **g** Illustration of different solving methods. Different parts of a neuron are assigned to multiple processing units in parallel methods (mid, right), shown with different colors. In the serial method (left), all compartments are computed with one unit. **h** Computational cost of three methods in **g** when solving equations of a pyramidal model with spines. **i** Run time of different methods on solving equations for 500 pyramidal models with spines. The run time indicates the time consumption of 1 s simulation (solving the equation 40,000 times with a time step of 0.025 ms). p-Hines parallel method in Cor-eNEURON (on GPU), Branch based branch-based parallel method (on GPU), DHS Dendritic hierarchical scheduling method (on GPU).

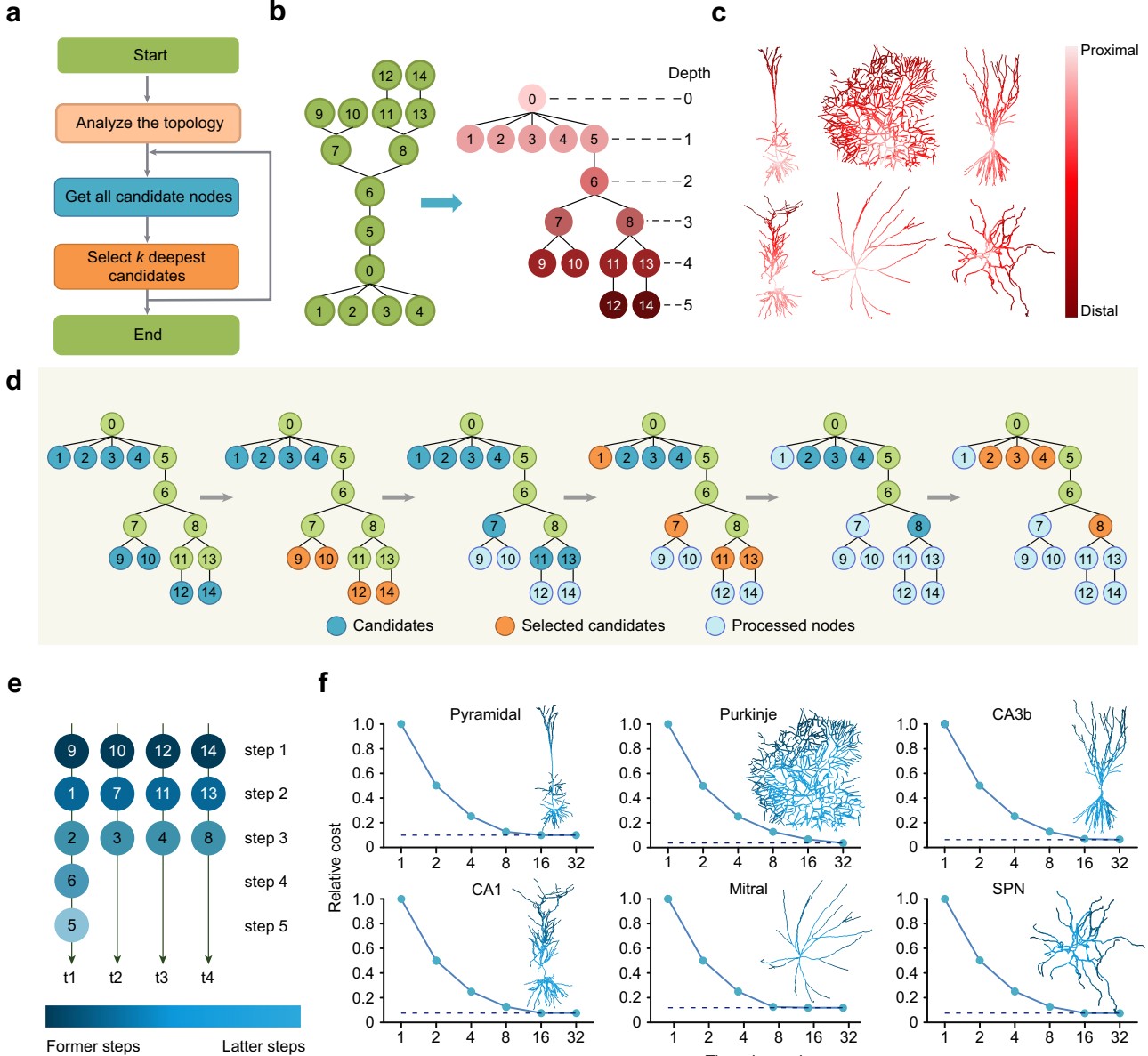

**Fig. 2 | Dendritic Hierarchical Scheduling (DHS) method significantly reduces the computational cost, i.e., computational steps in solving equations. a** DHS work flow. DHS processes *k* deepest candidate nodes each iteration. **b** Illustration of calculating node depth of a compartmental model. The model is first converted to a tree structure then the depth of each node is computed. Colors indicate different depth values. **c** Topology analysis on different neuron models. Six neurons with distinct morphologies are shown here. For each model, the soma is selected as the root of the tree so the depth of the node increases from the soma (0) to the distal dendrites. **d** Illustration of performing DHS on the model in **b** with four threads. Candidates: nodes that can be processed. Selected candidates: nodes that are picked by DHS, i.e., the *k* deepest candidates. Processed nodes: nodes that have been processed before. **e** Parallelization strategy obtained by DHS after the process in **d**. Each node is assigned to one of the four parallel threads. DHS reduces the steps of serial node processing from 14 to 5 by distributing nodes to multiple threads. **f** Relative cost, i.e., the proportion of the computational cost of DHS to that of the serial Hines method, when applying DHS with different numbers of threads on different types of models.

dependency in the Hines method (see Methods). Then to theoretically evaluate the run time, i.e., efficiency, of the serial and parallel computing methods, we introduce and formulate the concept of computational cost as the number of steps a method takes in solving equations (see Methods).

Based on the simulation accuracy and computational cost, we formulate the parallelization problem as a mathematical scheduling problem (see Methods). In simple terms, we view a single neuron as a tree with many nodes (compartments). For *k* parallel threads, we can compute at most *k* nodes at each step, but we need to ensure a node is computed only if all its children nodes have been processed; our goal is

to find a strategy with the minimum number of steps for the entire procedure.

To generate an optimal partition, we propose a method called Dendritic Hierarchical Scheduling (DHS) (theoretical proof is presented in the Methods). The key idea of DHS is to prioritize deep nodes (Fig. 2a), which results in a hierarchical schedule order. The DHS method includes two steps: analyzing dendritic topology and finding the best partition: (1) Given a detailed model, we first obtain its corresponding dependency tree and calculate the depth of each node (the depth of a node is the number of its ancestor nodes) on the tree (Fig. 2b, c). (2) After topology analysis, we search the candidates and

pick at most $k$ deepest candidate nodes (a node is a candidate only if all its children nodes have been processed). This procedure repeats until all nodes are processed (Fig. 2d).

Take a simplified model with 15 compartments as an example, using the serial computing Hines method, it takes 14 steps to process all nodes, while using DHS with four parallel units can partition its nodes into five subsets (Fig. 2d): {{9,10,12,14}, {1,7,11,13}, {2,3,4,8}, {6}, {5}}. Because nodes in the same subset can be processed in parallel, it takes only five steps to process all nodes using DHS (Fig. 2e).

Next, we apply the DHS method on six representative detailed neuron models (selected from ModelDB[39]) with different numbers of threads (Fig. 2f):, including cortical and hippocampal pyramidal neurons[40–42], cerebellar Purkinje neurons[43], striatal projection neurons (SPN)[44], and olfactory bulb mitral cells[45], covering the major principal neurons in sensory, cortical and subcortical areas. We then measured the computational cost. The relative computational cost here is defined by the proportion of the computational cost of DHS to that of the serial Hines method. The computational cost, i.e., the number of steps taken in solving equations, drops dramatically with increasing thread numbers. For example, with 16 threads, the computational cost of DHS is 7%-10% as compared to the serial Hines method. Intriguingly, the DHS method reaches the lower bounds of their computational cost for presented neurons when given 16 or even 8 parallel threads (Fig. 2f), suggesting adding more threads does not improve performance further because of the dependencies between compartments.

Together, we generate a DHS method that enables automated analysis of the dendritic topology and optimal partition for parallel computing. It is worth noting that DHS finds the optimal partition before the simulation starts, and no extra computation is needed to solve equations.

### Speeding up DHS by GPU memory boosting

DHS computes each neuron with multiple threads, which consumes a vast amount of threads when running neural network simulations. Graphics Processing Units (GPUs) consist of massive processing units (i.e., streaming processors, SPs, Fig. 3a, b) for parallel computing[46]. In theory, many SPs on the GPU should support efficient simulation for large-scale neural networks (Fig. 3c). However, we consistently observed that the efficiency of DHS significantly decreased when the network size grew, which might result from scattered data storage or extra memory access caused by loading and writing intermediate results (Fig. 3d, left).

We solve this problem by GPU memory boosting, a method to increase memory throughput by leveraging GPU's memory hierarchy and access mechanism. Based on the memory loading mechanism of GPU, successive threads loading aligned and successively-stored data lead to a high memory throughput compared to accessing scatter-stored data, which reduces memory throughput[46,47]. To achieve high throughput, we first align the computing orders of nodes and rearrange threads according to the number of nodes on them. Then we permute data storage in global memory, consistent with computing orders, i.e., nodes that are processed at the same step are stored successively in global memory. Moreover, we use GPU registers to store intermediate results, further strengthening memory throughput. The example shows that memory boosting takes only two memory transactions to load eight request data (Fig. 3d, right). Furthermore, experiments on multiple numbers of pyramidal neurons with spines and the typical neuron models (Fig. 3e, f; Supplementary Fig. 2) show that memory boosting achieves a 1.2-3.8 times speedup as compared to the naïve DHS.

To comprehensively test the performance of DHS with GPU memory boosting, we select six typical neuron models and evaluate the run time of solving cable equations on massive numbers of each model (Fig. 4). We examined DHS with four threads (DHS-4) and sixteen threads (DHS-16) for each neuron, respectively. Compared to the

GPU method in CoreNEURON, DHS-4 and DHS-16 can speed up about 5 and 15 times, respectively (Fig. 4a). Moreover, compared to the conventional serial Hines method in NEURON running with a single-thread of CPU, DHS speeds up the simulation by 2-3 orders of magnitude (Supplementary Fig. 3), while retaining the identical numerical accuracy in the presence of dense spines (Supplementary Figs. 4 and 8), active dendrites (Supplementary Fig. 7) and different segmentation strategies (Supplementary Fig. 7).

### DHS creates cell-type-specific optimal partitioning

To gain insights into the working mechanism of the DHS method, we visualized the partitioning process by mapping compartments to each thread (every color presents a single thread in Fig. 4b, c). The visualization shows that a single thread frequently switches among different branches (Fig. 4b, c). Interestingly, DHS generates aligned partitions in morphologically symmetric neurons such as the striatal projection neuron (SPN) and the Mitral cell (Fig. 4b, c). By contrast, it generates fragmented partitions of morphologically asymmetric neurons like the pyramidal neurons and Purkinje cell (Fig. 4b, c), indicating that DHS splits the neural tree at individual compartment scale (i.e., tree node) rather than branch scale. This cell-type-specific fine-grained partition enables DHS to fully exploit all available threads.

In summary, DHS and memory boosting generate a theoretically proven optimal solution for solving linear equations in parallel with unprecedented efficiency. Using this principle, we built the open-access DeepDendrite platform, which can be utilized by neuroscientists to implement models without any specific GPU programming knowledge. Below, we demonstrate how we can utilize DeepDendrite in neuroscience tasks. We also discuss the potential of the DeepDendrite framework for AI-related tasks in the Discussion section.

### DHS enables spine-level modelling

As dendritic spines receive most of the excitatory input to cortical and hippocampal pyramidal neurons, striatal projection neurons, etc., their morphologies and plasticity are crucial for regulating neuronal excitability[10,48–51]. However, spines are too small ( ~1 μm length) to be directly measured experimentally with regard to voltage-dependent processes. Thus, theoretical work is critical for the full understanding of the spine computations.

We can model a single spine with two compartments: the spine head where synapses are located and the spine neck that links the spine head to dendrites[52]. The theory predicts that the very thin spine neck (0.1-0.5 um in diameter) electronically isolates the spine head from its parent dendrite, thus compartmentalizing the signals generated at the spine head[53]. However, the detailed model with fully distributed spines on dendrites ("full-spine model") is computationally very expensive. A common compromising solution is to modify the capacitance and resistance of the membrane by a $F_{spine}$ factor[54], instead of modeling all spines explicitly. Here, the $F_{spine}$ factor aims at approximating the spine effect on the biophysical properties of the cell membrane[54].

Inspired by the previous work of Eyal et al. [51], we investigated how different spatial patterns of excitatory inputs formed on dendritic spines shape neuronal activities in a human pyramidal neuron model with explicitly modeled spines (Fig. 5a). Noticeably, Eyal et al. employed the $F_{spine}$ factor to incorporate spines into dendrites while only a few activated spines were explicitly attached to dendrites ("few-spine model" in Fig. 5a). The value of $F_{spine}$ in their model was computed from the dendritic area and spine area in the reconstructed data. Accordingly, we calculated the spine density from their reconstructed data to make our full-spine model more consistent with Eyal's few-spine model. With the spine density set to 1.3 μm$^{-1}$, the pyramidal neuron model contained about 25,000 spines without altering the model's original morphological and biophysical properties. Further, we repeated the previous experiment protocols with both full-spine and few-spine models. We use the same synaptic input as in Eyal's work

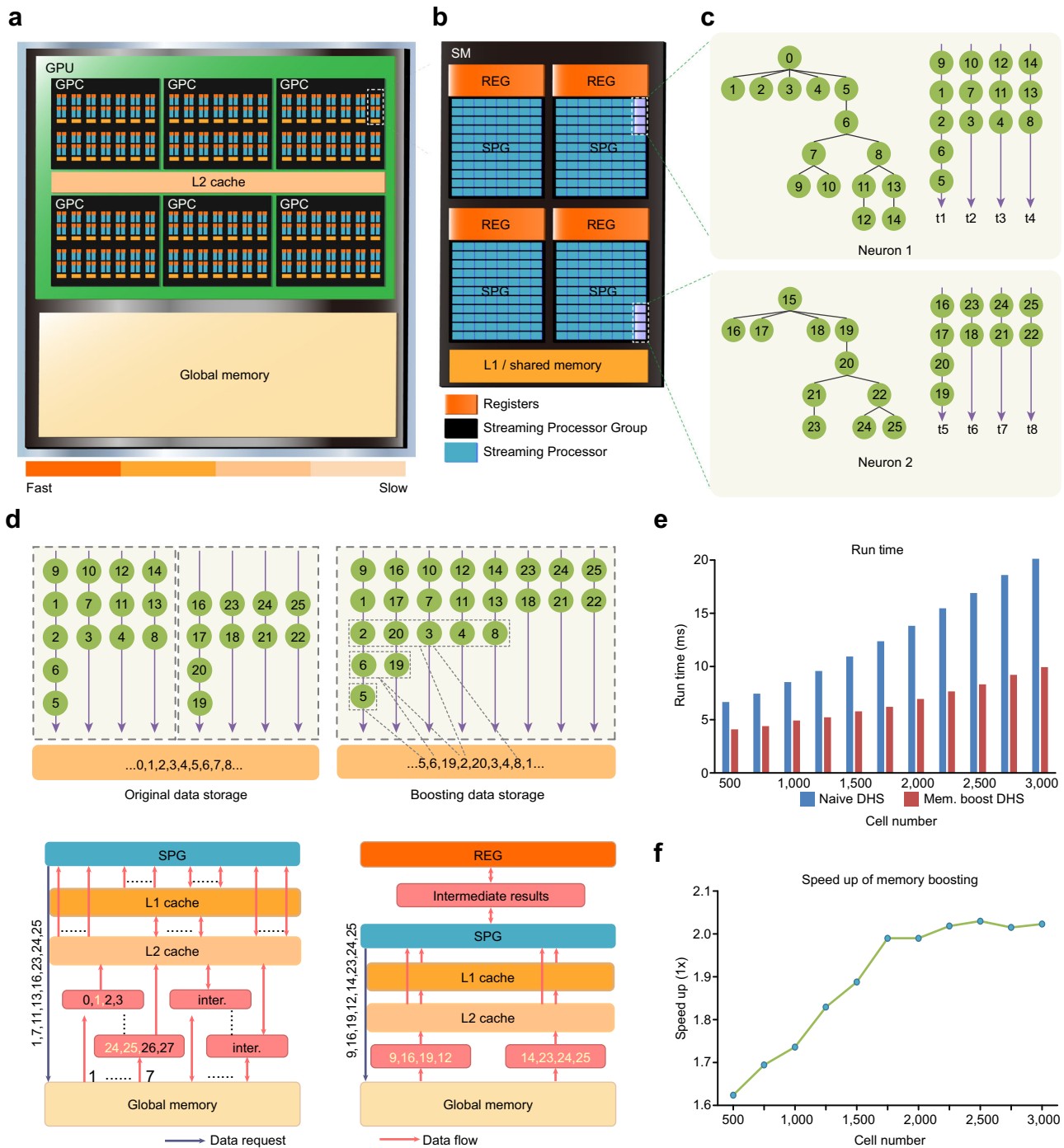

**Fig. 3 | GPU memory boosting further accelerates DHS. a** GPU architecture and its memory hierarchy. Each GPU contains massive processing units (stream processors). Different types of memory have different throughput. **b** Architecture of Streaming Multiprocessors (SMs). Each SM contains multiple streaming processors, registers, and L1 cache. **c** Applying DHS on two neurons, each with four threads. During simulation, each thread executes on one stream processor. **d** Memory optimization strategy on GPU. Top panel, thread assignment and data storage of DHS, before (left) and after (right) memory boosting. Bottom, an example of a single step in triangularization when simulating two neurons in **d**.

Processors send a data request to load data for each thread from global memory. Without memory boosting (left), it takes seven transactions to load all request data and some extra transactions for intermediate results. With memory boosting (right), it takes only two transactions to load all request data, registers are used for intermediate results, which further improve memory throughput. **e** Run time of DHS (32 threads each cell) with and without memory boosting on multiple layer 5 pyramidal models with spines. **f** Speed up of memory boosting on multiple layer 5 pyramidal models with spines. Memory boosting brings 1.6-2 times speedup.

but attach extra background noise to each sample. By comparing the somatic traces (Fig. 5b, c) and spike probability (Fig. 5d) in full-spine and few-spine models, we found that the full-spine model is much leakier than the few-spine model. In addition, the spike probability triggered by the activation of clustered spines appeared to be more nonlinear in the full-spine model (the solid blue line in Fig. 5d) than in the few-spine model (the dashed blue line in Fig. 5d). These results indicate that the conventional F-factor method may underestimate the impact of dense spine on the computations of dendritic excitability and nonlinearity.

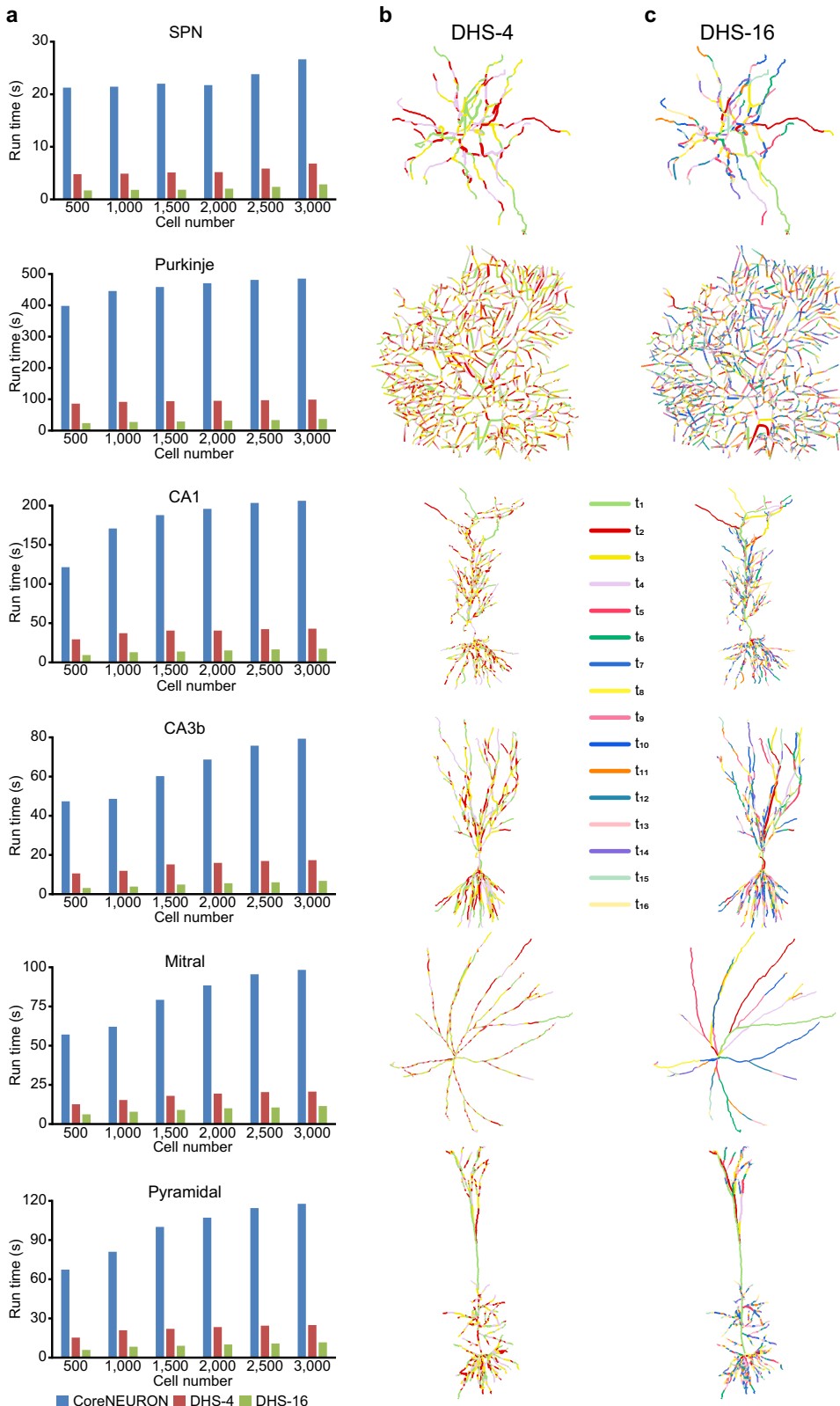

**Fig. 4 | DHS enables cell-type-specific finest partition. a** Run time of solving equations for a 1 s simulation on GPU (dt = 0.025 ms, 40,000 iterations in total). CoreNEURON: the parallel method used in CoreNEURON; DHS-4: DHS with four threads for each neuron; DHS-16: DHS with 16 threads for each neuron. **b, c** Visualization of the partition by DHS-4 and DHS-16, each color indicates a single thread. During computation, each thread switches among different branches.

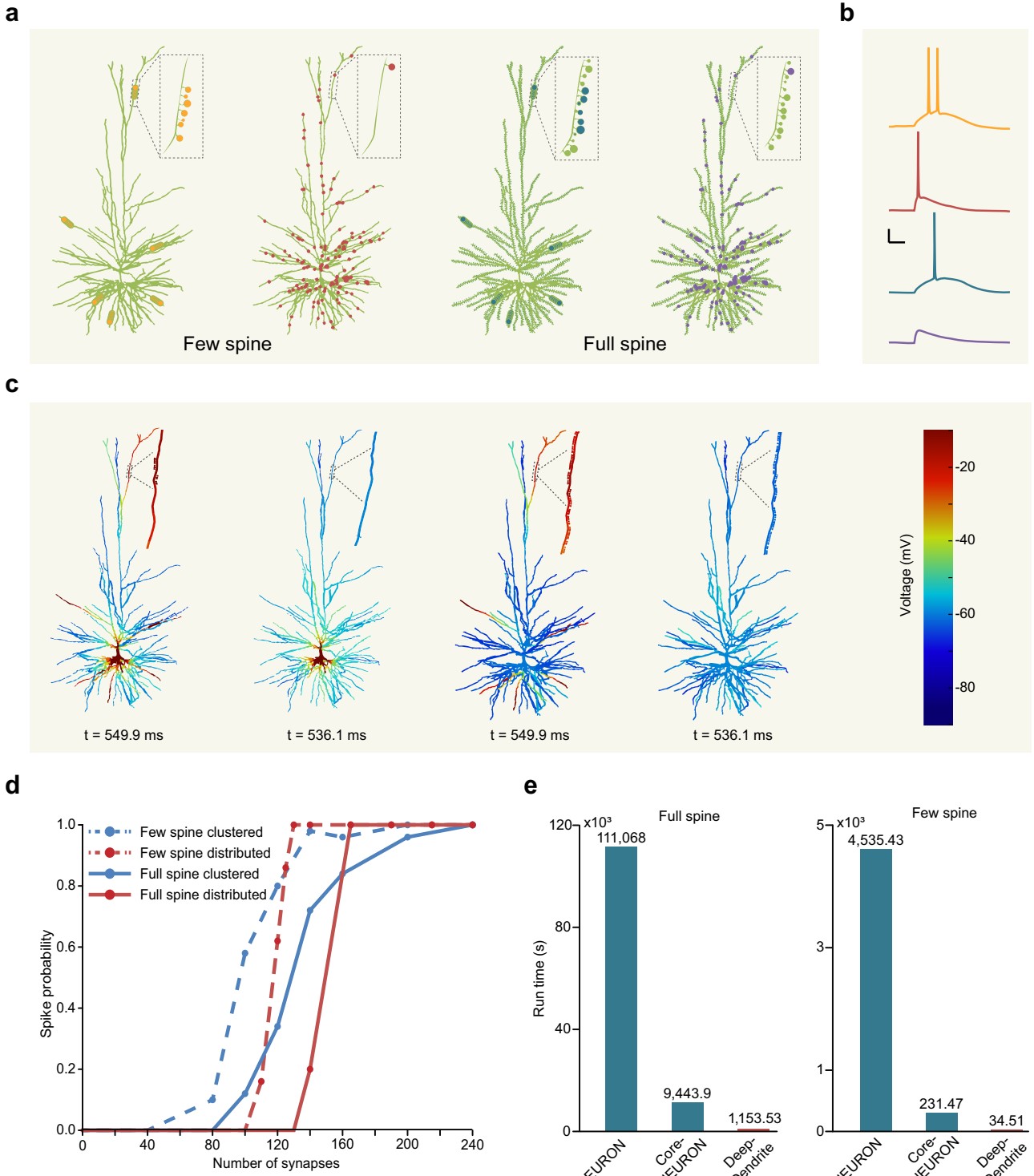

**Fig. 5 | DHS enables spine-level modeling. a** Experiment setup. We examine two major types of models: few-spine models and full-spine models. Few-spine models (two on the left) are the models that incorporated spine area globally into dendrites and only attach individual spines together with activated synapses. In full-spine models (two on the right), all spines are explicitly attached over whole dendrites. We explore the effects of clustered and randomly distributed synaptic inputs on the few-spine models and the full-spine models, respectively. **b** Somatic voltages recorded for cases in **a**. Colors of the voltage curves correspond to **a**, scale bar:

20 ms, 20 mV. **c** Color-coded voltages during the simulation in **b** at specific times. Colors indicate the magnitude of voltage. **d** Somatic spike probability as a function of the number of simultaneously activated synapses (as in Eyal et al.'s work) for four cases in **a**. Background noise is attached. **e** Run time of experiments in **d** with different simulation methods. NEURON: conventional NEURON simulator running on a single CPU core. CoreNEURON: CoreNEURON simulator on a single GPU. DeepDendrite: DeepDendrite on a single GPU.

In the DeepDendrite platform, both full-spine and few-spine models achieved 8 times speedup compared to CoreNEURON on the GPU platform and 100 times speedup compared to serial NEURON on the CPU platform (Fig. 5e; Supplementary Table 1) while keeping the identical simulation results (Supplementary Figs. 4 and 8). Therefore, the DHS method enables explorations of dendritic excitability under more realistic anatomic conditions.

## Discussion

In this work, we propose the DHS method to parallelize the computation of Hines method[55] and we mathematically demonstrate that the DHS provides an optimal solution without any loss of precision. Next, we implement DHS on the GPU hardware platform and use GPU memory boosting techniques to refine the DHS (Fig. 3). When simulating a large number of neurons with complex morphologies, DHS with memory boosting achieves a 15-fold speedup (Supplementary Table 1) as compared to the GPU method used in CoreNEURON and up to 1,500-fold speedup compared to serial Hines method in the CPU platform (Fig. 4; Supplementary Fig. 3 and Supplementary Table 1). Furthermore, we develop the GPU-based DeepDendrite framework by integrating DHS into CoreNEURON. Finally, as a demonstration of the capacity of DeepDendrite, we present a representative application: examine spine computations in a detailed pyramidal neuron model with 25,000 spines. Further in this section, we elaborate on how we have expanded the DeepDendrite framework to enable efficient training of biophysically detailed neural networks. To explore the hypothesis that dendrites improve robustness against adversarial attacks[56], we train our network on typical image classification tasks. We show that DeepDendrite can support both neuroscience simulations and AI-related detailed neural network tasks with unprecedented speed, therefore significantly promoting detailed neuroscience simulations and potentially for future AI explorations.

Decades of efforts have been invested in speeding up the Hines method with parallel methods. Early work mainly focuses on network-level parallelization. In network simulations, each cell independently solves its corresponding linear equations with the Hines method. Network-level parallel methods distribute a network on multiple threads and parallelize the computation of each cell group with each thread[57,58]. With network-level methods, we can simulate detailed networks on clusters or supercomputers[59]. In recent years, GPU has been used for detailed network simulation. Because the GPU contains massive computing units, one thread is usually assigned one cell rather than a cell group[35,60,61]. With further optimization, GPU-based methods achieve much higher efficiency in network simulation. However, the computation inside the cells is still serial in network-level methods, so they still cannot deal with the problem when the "Hines matrix" of each cell scales large.

Cellular-level parallel methods further parallelize the computation inside each cell. The main idea of cellular-level parallel methods is to split each cell into several sub-blocks and parallelize the computation of those sub-blocks[27,28]. However, typical cellular-level methods (e.g., the "multi-split" method[28]) pay less attention to the parallelization strategy. The lack of a fine parallelization strategy results in unsatisfactory performance. To achieve higher efficiency, some studies try to obtain finer-grained parallelization by introducing extra computation operations[29,38,62] or making approximations on some crucial compartments, while solving linear equations[63,64]. These finer-grained parallelization strategies can get higher efficiency but lack sufficient numerical accuracy as in the original Hines method.

Unlike previous methods, DHS adopts the finest-grained parallelization strategy, i.e., compartment-level parallelization. By modeling the problem of "how to parallelize" as a combinatorial optimization problem, DHS provides an optimal compartment-level parallelization strategy. Moreover, DHS does not introduce any extra operation or value approximation, so it achieves the lowest computational cost and

retains sufficient numerical accuracy as in the original Hines method at the same time.

Dendritic spines are the most abundant microstructures in the brain for projection neurons in the cortex, hippocampus, cerebellum, and basal ganglia. As spines receive most of the excitatory inputs in the central nervous system, electrical signals generated by spines are the main driving force for large-scale neuronal activities in the forebrain and cerebellum[10,11]. The structure of the spine, with an enlarged spine head and a very thin spine neck−leads to surprisingly high input impedance at the spine head, which could be up to 500 MΩ, combining experimental data and the detailed compartment modeling approach[48,65]. Due to such high input impedance, a single synaptic input can evoke a "gigantic" EPSP (∼20 mV) at the spine-head level[48,66], thereby boosting NMDA currents and ion channel currents in the spine[11]. However, in the classic single detailed compartment models, all spines are replaced by the $F$ coefficient modifying the dendritic cable geometries[54]. This approach may compensate for the leak currents and capacitance currents for spines. Still, it cannot reproduce the high input impedance at the spine head, which may weaken excitatory synaptic inputs, particularly NMDA currents, thereby reducing the nonlinearity in the neuron's input-output curve. Our modeling results are in line with this interpretation.

On the other hand, the spine's electrical compartmentalization is always accompanied by the biochemical compartmentalization[8,52,67], resulting in a drastic increase of internal $[Ca^{2+}]_i$ within the spine and a cascade of molecular processes involving synaptic plasticity of importance for learning and memory. Intriguingly, the biochemical process triggered by learning, in turn, remodels the spine's morphology, enlarging (or shrinking) the spine head, or elongating (or shortening) the spine neck, which significantly alters the spine's electrical capacity[67–70]. Such experience-dependent changes in spine morphology also referred to as "structural plasticity", have been widely observed in the visual cortex[71,72], somatosensory cortex[73,74], motor cortex[75], hippocampus[9], and the basal ganglia[76] in vivo. They play a critical role in motor and spatial learning as well as memory formation. However, due to the computational costs, nearly all detailed network models exploit the "F-factor" approach to replace actual spines, and are thus unable to explore the spine functions at the system level. By taking advantage of our framework and the GPU platform, we can run a few thousand detailed neurons models, each with tens of thousands of spines on a single GPU, while maintaining ∼100 times faster than the traditional serial method on a single CPU (Fig. 5e). Therefore, it enables us to explore of structural plasticity in large-scale circuit models across diverse brain regions.

Another critical issue is how to link dendrites to brain functions at the systems/network level. It has been well established that dendrites can perform comprehensive computations on synaptic inputs due to enriched ion channels and local biophysical membrane properties[5–7]. For example, cortical pyramidal neurons can carry out sublinear synaptic integration at the proximal dendrite but progressively shift to supralinear integration at the distal dendrite[77]. Moreover, distal dendrites can produce regenerative events such as dendritic sodium spikes, calcium spikes, and NMDA spikes/plateau potentials[6,78]. Such dendritic events are widely observed in mice[6] or even human cortical neurons[79] in vitro, which may offer various logical operations[6,79] or gating functions[80,81]. Recently, in vivo recordings in awake or behaving mice provide strong evidence that dendritic spikes/plateau potentials are crucial for orientation selectivity in the visual cortex[82], sensory-motor integration in the whisker system[83,84], and spatial navigation in the hippocampal CA1 region[85].

To establish the causal link between dendrites and animal (including human) patterns of behavior, large-scale biophysically detailed neural circuit models are a powerful computational tool to realize this mission. However, running a large-scale detailed circuit model of 10,000-100,000 neurons generally requires the computing

power of supercomputers. It is even more challenging to optimize such models for in vivo data, as it needs iterative simulations of the models. The DeepDendrite framework can directly support many state-of-the-art large-scale circuit models[86–88], which were initially developed based on NEURON. Moreover, using our framework, a single GPU card such as Tesla A100 could easily support the operation of detailed circuit models of up to 10,000 neurons, thereby providing carbon-efficient and affordable plans for ordinary labs to develop and optimize their own large-scale detailed models.

Recent works on unraveling the dendritic roles in task-specific learning have achieved remarkable results in two directions, i.e., solving challenging tasks such as image classification dataset ImageNet with simplified dendritic networks[20], and exploring full learning potentials on more realistic neuron[21,22]. However, there lies a trade-off between model size and biological detail, as the increase in network scale is often sacrificed for neuron-level complexity[19,20,89]. Moreover, more detailed neuron models are less mathematically tractable and computationally expensive[21].

There has also been progress in the role of active dendrites in ANNs for computer vision tasks. Iyer et al.[90]. proposed a novel ANN architecture with active dendrites, demonstrating competitive results in multi-task and continual learning. Jones and Kording[91] used a binary tree to approximate dendrite branching and provided valuable insights into the influence of tree structure on single neurons' computational capacity. Bird et al.[92]. proposed a dendritic normalization rule based on biophysical behavior, offering an interesting perspective on the contribution of dendritic arbor structure to computation. While these studies offer valuable insights, they primarily rely on abstractions derived from spatially extended neurons, and do not fully exploit the detailed biological properties and spatial information of dendrites. Further investigation is needed to unveil the potential of leveraging more realistic neuron models for understanding the shared mechanisms underlying brain computation and deep learning.

In response to these challenges, we developed DeepDendrite, a tool that uses the Dendritic Hierarchical Scheduling (DHS) method to significantly reduce computational costs and incorporates an I/O module and a learning module to handle large datasets. With DeepDendrite, we successfully implemented a three-layer hybrid neural network, the Human Pyramidal Cell Network (HPC-Net) (Fig. 6a, b). This network demonstrated efficient training capabilities in image classification tasks, achieving approximately 25 times speedup compared to training on a traditional CPU-based platform (Fig. 6f; Supplementary Table 1).

Additionally, it is widely recognized that the performance of Artificial Neural Networks (ANNs) can be undermined by adversarial attacks[93]−intentionally engineered perturbations devised to mislead ANNs. Intriguingly, an existing hypothesis suggests that dendrites and synapses may innately defend against such attacks[56]. Our experimental results utilizing HPC-Net lend support to this hypothesis, as we observed that networks endowed with detailed dendritic structures demonstrated some increased resilience to transfer adversarial attacks[94] compared to standard ANNs, as evident in MNIST[95] and Fashion-MNIST[96] datasets (Fig. 6d, e). This evidence implies that the inherent biophysical properties of dendrites could be pivotal in augmenting the robustness of ANNs against adversarial interference. Nonetheless, it is essential to conduct further studies to validate these findings using more challenging datasets such as ImageNet[97].

In conclusion, DeepDendrite has shown remarkable potential in image classification tasks, opening up a world of exciting future directions and possibilities. To further advance DeepDendrite and the application of biologically detailed dendritic models in AI tasks, we may focus on developing multi-GPU systems and exploring applications in other domains, such as Natural Language Processing (NLP), where dendritic filtering properties align well with the inherently noisy and ambiguous nature of human language. Challenges include

testing scalability in larger-scale problems, understanding performance across various tasks and domains, and addressing the computational complexity introduced by novel biological principles, such as active dendrites. By overcoming these limitations, we can further advance the understanding and capabilities of biophysically detailed dendritic neural networks, potentially uncovering new advantages, enhancing their robustness against adversarial attacks and noisy inputs, and ultimately bridging the gap between neuroscience and modern AI.

## Methods
### Simulation with DHS
CoreNEURON[35] simulator (https://github.com/BlueBrain/CoreNeuron) uses the NEURON[25] architecture and is optimized for both memory usage and computational speed. We implement our Dendritic Hierarchical Scheduling (DHS) method in the CoreNEURON environment by modifying its source code. All models that can be simulated on GPU with CoreNEURON can also be simulated with DHS by executing the following command:

coreneuron_exec -d /path/to/models -e time --cell-permute 3 --cell-nthread 16 --gpu

The usage options are as in Table 1.

### Accuracy of the simulation using cellular-level parallel computation
To ensure the accuracy of the simulation, we first need to define the correctness of a cellular-level parallel algorithm to judge whether it will generate identical solutions compared with the proven correct serial methods, like the Hines method used in the NEURON simulation platform. Based on the theories in parallel computing[34], a parallel algorithm will yield an identical result as its corresponding serial algorithm, if and only if the data process order in the parallel algorithm is consistent with data dependency in the serial method. The Hines method has two symmetrical phases: triangularization and back-substitution. By analyzing the serial computing Hines method[55], we find that its data dependency can be formulated as a tree structure, where the nodes on the tree represent the compartments of the detailed neuron model. In the triangularization process, the value of each node depends on its children nodes. In contrast, during the back-substitution process, the value of each node is dependent on its parent node (Fig. 1d). Thus, we can compute nodes on different branches in parallel as their values are not dependent.

Based on the data dependency of the serial computing Hines method, we propose three conditions to make sure a parallel method will yield identical solutions as the serial computing Hines method: (1) The tree morphology and initial values of all nodes are identical to those in the serial computing Hines method; (2) In the triangularization phase, a node can be processed if and only if all its children nodes are already processed; (3) In the back-substitution phase, a node can be processed only if its parent node is already processed. Once a parallel computing method satisfies these three conditions, it will produce identical solutions as the serial computing method.

### Computational cost of cellular-level parallel computing method
To theoretically evaluate the run time, i.e., efficiency, of the serial and parallel computing methods, we introduce and formulate the concept of computational cost as follows: given a tree $T$ and $k$ threads (basic computational units) to perform triangularization, parallel triangularization equals to divide the node set $V$ of $T$ into $n$ subsets, i.e., $V = \{V_1, V_2, \ldots V_n\}$ where the size of each subset $|V_i| \leq k$, i.e., at most $k$ nodes can be processed each step since there are only $k$ threads. The process of the triangularization phase follows the order: $V_1 \rightarrow V_2 \rightarrow \ldots \rightarrow V_n$, and nodes in the same subset $V_i$ can be processed in parallel. So, we define $|V|$ (the size of set $V$, i.e., $n$ here) as the computational cost of the parallel computing method. In short, we define the computational cost

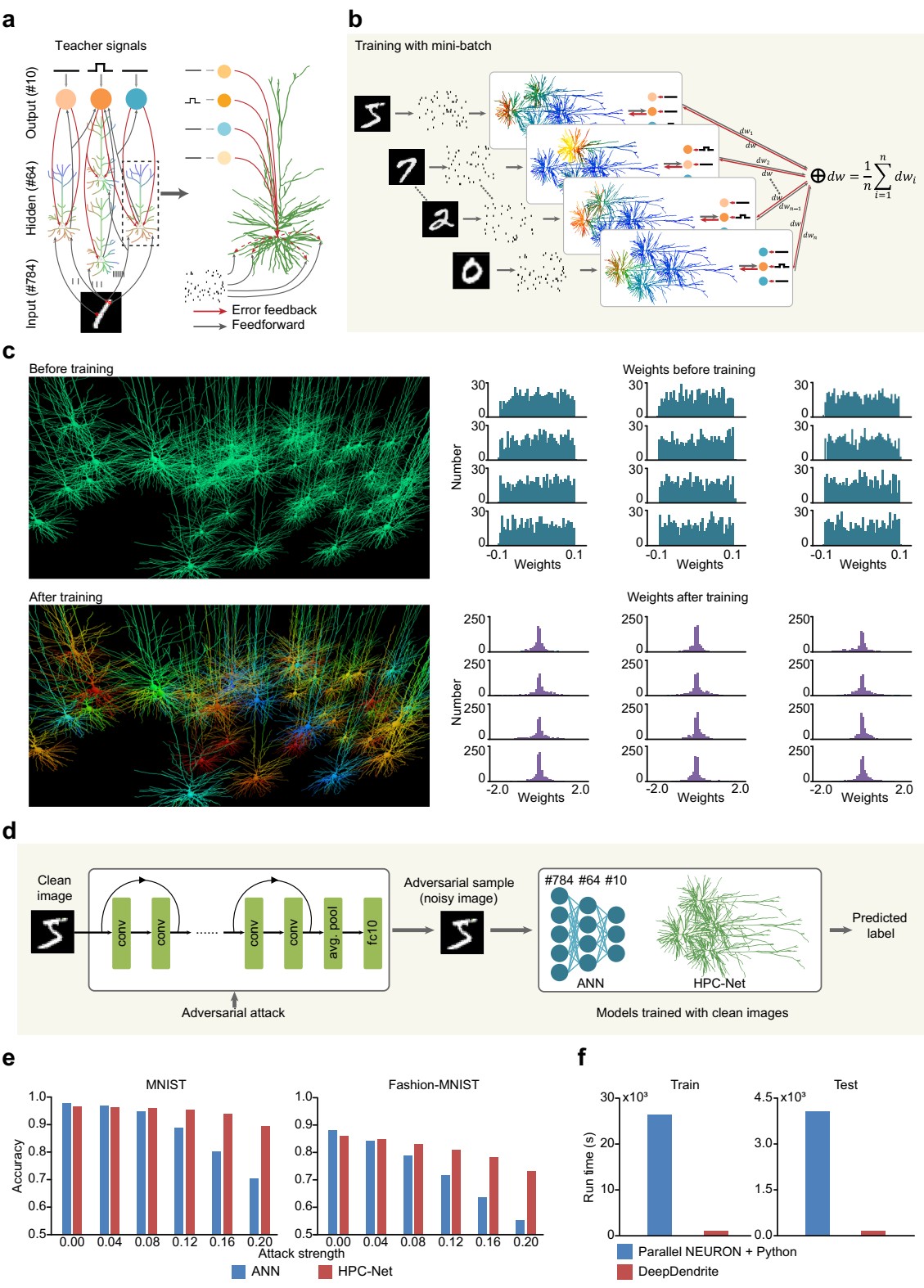

of a parallel method as the number of steps it takes in the triangular-ization phase. Because the back-substitution is symmetrical with tri-angularization, the total cost of the entire solving equation phase is twice that of the triangularization phase.

## Mathematical scheduling problem

Based on the simulation accuracy and computational cost, we for-mulate the parallelization problem as a mathematical scheduling problem:

Given a tree $T = \{V, E\}$ and a positive integer $k$, where $V$ is the node-set and $E$ is the edge set. Define partition $P(V) = \{V_1, V_2, \dots V_n\}$, $|V_i| \le k$, $1 \le i \le n$, where $|V_i|$ indicates the cardinal number of subset $V_i$, i.e., the number of nodes in $V_i$, and for each node $v \in V_i$, all its children nodes $\{c \mid c \in \text{children}(v)\}$ must in a previous subset $V_j$, where $1 \le j < i$. Our goal is to find an optimal partition $P^*(V)$ whose computational cost $|P^*(V)|$ is minimal.

Here subset $V_i$ consists of all nodes that will be computed at $i$-th step (Fig. 2e), so $|V_i| \le k$ indicates that we can compute $k$ nodes each

**Fig. 6 | DeepDendrite enables learning with detailed neural networks. a** The illustration of the Human Pyramidal Cell Network (HPC-Net) for image classification. Images are transformed to spike trains and fed into the network model. Learning is triggered by error signals propagated from soma to dendrites. **b** Training with mini-batch. Multiple networks are simulated simultaneously with different images as inputs. The total weight updates $\Delta W$ are computed as the average of $\Delta W_i$ from each network. **c** Comparison of the HPC-Net before and after training. Left, the visualization of hidden neuron responses to a specific input before (top) and after (bottom) training. Right, hidden layer weights (from input to hidden layer) distribution before (top) and after (bottom) training. **d** Workflow of the transfer adversarial attack experiment. We first generate adversarial samples of the test set on a 20-layer ResNet. Then use these adversarial samples (noisy images) to test the classification accuracy of models trained with clean images. **e** Prediction accuracy of each model on adversarial samples after training 30 epochs on MNIST (left) and Fashion-MNIST (right) datasets. **f** Run time of training and testing for the HPC-Net. The batch size is set to 16. Left, run time of training one epoch. Right, run time of testing. Parallel NEURON + Python: training and testing on a single CPU with multiple cores, using 40-process-parallel NEURON to simulate the HPC-Net and extra Python code to support mini-batch training. DeepDendrite: training and testing the HPC-Net on a single GPU with DeepDendrite.

## Table 1 | Usage options for DHS-embedded CoreNEURON

| | |
|---|---|
| -d | Path containing the model data |
| -e | Simulation time (ms) |
| --cell-permute | Strategy for optimizing simulation: 1 and 2 for original strategies in CoreNEURON, **3 for DHS method** |
| --cell-nthread | Number of threads used for each cell |
| --gpu | Simulate on GPU |

step at most because the number of available threads is $k$. The restriction "for each node $v \in V_i$, all its children nodes $\{c \mid c \in \text{children}(v)\}$ must in a previous subset $V_j$, where $1 \le j < i$" indicates that node $v$ can be processed only if all its child nodes are processed.

### DHS implementation

We aim to find an optimal way to parallelize the computation of solving linear equations for each neuron model by solving the mathematical scheduling problem above. To get the optimal partition, DHS first analyzes the topology and calculates the depth $d(v)$ for all nodes $v \in V$. Then, the following two steps will be executed iteratively until every node $v \in V$ is assigned to a subset: (1) find all candidate nodes and put these nodes into candidate set $Q$. A node is a candidate only if all its child nodes have been processed or it does not have any child nodes. (2) if $|Q| \le k$, i.e., the number of candidate nodes is smaller or equivalent to the number of available threads, remove all nodes in $Q$ and put them into $V_i$, otherwise, remove $k$ deepest nodes from $Q$ and add them to subset $V_i$. Label these nodes as processed nodes (Fig. 2d). After filling in subset $V_i$, go to step (1) to fill in the next subset $V_{i+1}$.

### Correctness proof for DHS

After applying DHS to a neural tree $T = \{V, E\}$, we get a partition $P(V) = \{V_1, V_2, \dots V_n\}$, $|V_i| \le k$, $1 \le i \le n$. Nodes in the same subset $V_i$ will be computed in parallel, taking $n$ steps to perform triangularization and back-substitution, respectively. We then demonstrate that the reordering of the computation in DHS will result in a result identical to the serial Hines method.

The partition $P(V)$ obtained from DHS decides the computation order of all nodes in a neural tree. Below we demonstrate that the computation order determined by $P(V)$ satisfies the correctness conditions. $P(V)$ is obtained from the given neural tree $T$. Operations in DHS do not modify the tree topology and values of tree nodes (corresponding values in the linear equations), so the tree morphology and initial values of all nodes are not changed, which satisfies condition 1: the tree morphology and initial values of all nodes are identical to those in serial Hines method. In triangularization, nodes are processed from subset $V_1$ to $V_n$. As shown in the implementation of DHS, all nodes in subset $V_i$ are selected from the candidate set $Q$, and a node can be put into $Q$ only if all its child nodes have been processed. Thus the child nodes of all nodes in $V_i$ are in $\{V_1, V_2, \dots V_{i-1}\}$, meaning that a node is only computed after all its children have been processed, which satisfies condition 2: in triangularization, a node can be processed if and only if all its child nodes are already processed. In back-substitution, the computation order is the opposite of that in triangularization, i.e., from $V_n$ to $V_1$. As shown before, the child nodes of all nodes in $V_i$ are in $\{V_1, V_2, \dots V_{i-1}\}$, so parent nodes of nodes in $V_i$ are in $\{V_{i+1}, V_{i+2}, \dots V_n\}$, which satisfies condition 3: in back-substitution, a node can be processed only if its parent node is already processed.

### Optimality proof for DHS

The idea of the proof is that if there is another optimal solution, it can be transformed into our DHS solution without increasing the number of steps the algorithm requires, thus indicating that the DHS solution is optimal.

For each subset $V_i$ in $P(V)$, DHS moves $k$ (thread number) deepest nodes from the corresponding candidate set $Q_i$ to $V_i$. If the number of nodes in $Q_i$ is smaller than $k$, move all nodes from $Q_i$ to $V_i$. To simplify, we introduce $D_i$, indicating the depth sum of $k$ deepest nodes in $Q_i$. All subsets in $P(V)$ satisfy the max-depth criteria (Supplementary Fig. 6a): $\sum_{v_i \in V_i} d(v_i) = D_i$. We then prove that selecting the deepest nodes in each iteration makes $P(V)$ an optimal partition. If there exists an optimal partition $P^*(V) = \{V_1^*, V_2^*, \dots V_s^*\}$ containing subsets that do not satisfy the max-depth criteria, we can modify the subsets in $P^*(V)$ so that all subsets consist of the deepest nodes from $Q$ and the number of subsets ($|P^*(V)|$) remain the same after modification.

Without any loss of generalization, we start from the first subset $V_i^*$ not satisfying the criteria, i.e., $\sum_{v_i^* \in V_i^*} d(v_i^*) < D_i$. There are two possible cases that will make $V_i^*$ not satisfy the max-depth criteria: (1) $|V_i^*| < k$ and there exist some valid nodes in $Q_i$ that are not put to $V_i^*$; (2) $|V_i^*| = k$ but nodes in $V_i^*$ are not the $k$ deepest nodes in $Q_i$.

For case (1), because some candidate nodes are not put to $V_i^*$, these nodes must be in the subsequent subsets. As $|V_i^*| < k$, we can move the corresponding nodes from the subsequent subsets to $V_i^*$, which will not increase the number of subsets and make $V_i^*$ satisfy the criteria (Supplementary Fig. 6b, top). For case (2), $|V_i^*| = k$, these deeper nodes that are not moved from the candidate set $Q_i$ into $V_i^*$ must be added to subsequent subsets (Supplementary Fig. 6b, bottom). These deeper nodes can be moved from subsequent subsets to $V_i^*$ through the following method. Assume that after filling $V_i^*$, $v$ is picked and one of the $k$-th deepest nodes $v'$ is still in $Q_i$, thus $v'$ will be put into a subsequent subset $V_j$ ($j > i$). We first move $v$ from $V_i^*$ to $V_{i+1}$, then modify subset $V_{i+1}$ as follows: if $|V_{i+1}| \le k$ and none of the nodes in $V_{i+1}$ is the parent of node $v$, stop modifying the latter subsets. Otherwise, modify $V_{i+1}$ as follows (Supplementary Fig. 6c): if the parent node of $v$ is in $V_{i+1}$, move this parent node to $V_{i+2}$; else move the node with minimum depth from $V_{i+1}$ to $V_{i+2}$. After adjusting $V_i^*$, modify subsequent subsets $V_{i+1}$, $V_{i+2}$, ... $V_{j-1}$ with the same strategy. Finally, move $v'$ from $V_j$ to $V_i^*$.

With the modification strategy described above, we can replace all shallower nodes in $V_i^*$ with the $k$-th deepest node in $Q_i$ and keep the number of subsets, i.e., $|P^*(V)|$ the same after modification. We can modify the nodes with the same strategy for all subsets in $P^*(V)$ that do not contain the deepest nodes. Finally, all subsets $V_i^* \in P^*(V)$ can satisfy the max-depth criteria, and $|P^*(V)|$ does not change after modifying.

In conclusion, DHS generates a partition $P(V)$, and all subsets $V_i \in P(V)$ satisfy the max-depth condition: $\sum_{v_i \in V_i} d(v_i) = D_i$. For any other optimal partition $P'(V)$ we can modify its subsets to make its structure the same as $P(V)$, i.e., each subset consists of the deepest nodes in the candidate set, and keep $|P'(V)|$ the same after modification. So, the partition $P(V)$ obtained from DHS is one of the optimal partitions.

## GPU implementation and memory boosting

To achieve high memory throughput, GPU utilizes the memory hierarchy of (1) global memory, (2) cache, (3) register, where global memory has large capacity but low throughput, while registers have low capacity but high throughput. We aim to boost memory throughput by leveraging the memory hierarchy of GPU.

GPU employs SIMT (Single-Instruction, Multiple-Thread) architecture. Warps are the basic scheduling units on GPU (a warp is a group of 32 parallel threads). A warp executes the same instruction with different data for different threads[46]. Correctly ordering the nodes is essential for this batching of computation in warps, to make sure DHS obtains identical results as the serial Hines method. When implementing DHS on GPU, we first group all cells into multiple warps based on their morphologies. Cells with similar morphologies are grouped in the same warp. We then apply DHS on all neurons, assigning the compartments of each neuron to multiple threads. Because neurons are grouped into warps, the threads for the same neuron are in the same warp. Therefore, the intrinsic synchronization in warps keeps the computation order consistent with the data dependency of the serial Hines method. Finally, threads in each warp are aligned and rearranged according to the number of compartments.

When a warp loads pre-aligned and successively-stored data from global memory, it can make full use of the cache, which leads to high memory throughput, while accessing scatter-stored data would reduce memory throughput. After compartments assignment and threads rearrangement, we permute data in global memory to make it consistent with computing orders so that warps can load successively-stored data when executing the program. Moreover, we put those necessary temporary variables into registers rather than global memory. Registers have the highest memory throughput, so the use of registers further accelerates DHS.

## Full-spine and few-spine biophysical models

We used the published human pyramidal neuron[51]. The membrane capacitance $c_m = 0.44\,\mu\text{F cm}^{-2}$, membrane resistance $r_m = 48{,}300\,\Omega\,\text{cm}^2$, and axial resistivity $r_a = 261.97\,\Omega\,\text{cm}$. In this model, all dendrites were modeled as passive cables while somas were active. The leak reversal potential $E_l = -83.1\,\text{mV}$. Ion channels such as $Na^+$ and $K^+$ were inserted on soma and initial axon, and their reversal potentials were $E_{Na} = 67.6\,\text{mV}$, $E_K = -102\,\text{mV}$ respectively. All these specific parameters were set the same as in the model of Eyal, et al. [51], for more details please refer to the published model (ModelDB, access No. 238347).

In the few-spine model, the membrane capacitance and maximum leak conductance of the dendritic cables 60 μm away from soma were multiplied by a $F_{spine}$ factor to approximate dendritic spines. In this model, $F_{spine}$ was set to 1.9. Only the spines that receive synaptic inputs were explicitly attached to dendrites.

In the full-spine model, all spines were explicitly attached to dendrites. We calculated the spine density with the reconstructed neuron in Eyal, et al. [51]. The spine density was set to 1.3 μm⁻¹, and each cell contained 24994 spines on dendrites 60 μm away from the soma.

The morphologies and biophysical mechanisms of spines were the same in few-spine and full-spine models. The length of the spine neck $L_{neck} = 1.35\,\mu\text{m}$ and the diameter $D_{neck} = 0.25\,\mu\text{m}$, whereas the length and diameter of the spine head were 0.944 μm, i.e., the spine head area was set to 2.8 μm². Both spine neck and spine head were modeled as passive cables, with the reversal potential $E_l = -86\,\text{mV}$. The

specific membrane capacitance, membrane resistance, and axial resistivity were the same as those for dendrites.

## Synaptic inputs

We investigated neuronal excitability for both distributed and clustered synaptic inputs. All activated synapses were attached to the terminal of the spine head. For distributed inputs, all activated synapses were randomly distributed on all dendrites. For clustered inputs, each cluster consisted of 20 activated synapses that were uniformly distributed on a single randomly-selected compartment. All synapses were activated simultaneously during the simulation.

AMPA-based and NMDA-based synaptic currents were simulated as in Eyal et al.'s work. AMPA conductance was modeled as a double-exponential function and NMDA conduction as a voltage-dependent double-exponential function. For the AMPA model, the specific $\tau_{rise}$ and $\tau_{decay}$ were set to 0.3 and 1.8 ms. For the NMDA model, $\tau_{rise}$ and $\tau_{decay}$ were set to 8.019 and 34.9884 ms, respectively. The maximum conductance of AMPA and NMDA were 0.73 nS and 1.31 nS.

## Background noise

We attached background noise to each cell to simulate a more realistic environment. Noise patterns were implemented as Poisson spike trains with a constant rate of 1.0 Hz. Each pattern started at $t_{start} = 10\,\text{ms}$ and lasted until the end of the simulation. We generated 400 noise spike trains for each cell and attached them to randomly-selected synapses. The model and specific parameters of synaptic currents were the same as described in **Synaptic Inputs**, except that the maximum conductance of NMDA was uniformly distributed from 1.57 to 3.275, resulting in a higher AMPA to NMDA ratio.

## Exploring neuronal excitability

We investigated the spike probability when multiple synapses were activated simultaneously. For distributed inputs, we tested 14 cases, from 0 to 240 activated synapses. For clustered inputs, we tested 9 cases in total, activating from 0 to 12 clusters respectively. Each cluster consisted of 20 synapses. For each case in both distributed and clustered inputs, we calculated the spike probability with 50 random samples. Spike probability was defined as the ratio of the number of neurons fired to the total number of samples. All 1150 samples were simulated simultaneously on our DeepDendrite platform, reducing the simulation time from days to minutes.

## Performing AI tasks with the DeepDendrite platform

Conventional detailed neuron simulators lack two functionalities important to modern AI tasks: (1) alternately performing simulations and weight updates without heavy reinitialization and (2) simultaneously processing multiple stimuli samples in a batch-like manner. Here we present the DeepDendrite platform, which supports both biophysical simulating and performing deep learning tasks with detailed dendritic models.

DeepDendrite consists of three modules (Supplementary Fig. 5): (1) an I/O module; (2) a DHS-based simulating module; (3) a learning module. When training a biophysically detailed model to perform learning tasks, users first define the learning rule, then feed all training samples to the detailed model for learning. In each step during training, the I/O module picks a specific stimulus and its corresponding teacher signal (if necessary) from all training samples and attaches the stimulus to the network model. Then, the DHS-based simulating module initializes the model and starts the simulation. After simulation, the learning module updates all synaptic weights according to the difference between model responses and teacher signals. After training, the learned model can achieve performance comparable to ANN. The testing phase is similar to training, except that all synaptic weights are fixed.

## HPC-Net model

Image classification is a typical task in the field of AI. In this task, a model should learn to recognize the content in a given image and output the corresponding label. Here we present the HPC-Net, a network consisting of detailed human pyramidal neuron models that can learn to perform image classification tasks by utilizing the DeepDendrite platform.

HPC-Net has three layers, i.e., an input layer, a hidden layer, and an output layer. The neurons in the input layer receive spike trains converted from images as their input. Hidden layer neurons receive the output of input layer neurons and deliver responses to neurons in the output layer. The responses of the output layer neurons are taken as the final output of HPC-Net. Neurons between adjacent layers are fully connected.

For each image stimulus, we first convert each normalized pixel to a homogeneous spike train. For pixel with coordinates $(x, y)$ in the image, the corresponding spike train has a constant interspike interval $\tau_{\text{ISI}}(x, y)$ (in ms) which is determined by the pixel value $p(x, y)$ as shown in Eq. (1).

$$\tau_{\text{ISI}}(x, y) = \frac{5}{p(x, y) + 0.01} \tag{1}$$

In our experiment, the simulation for each stimulus lasted 50 ms. All spike trains started at $9 + \tau_{\text{ISI}}$ ms and lasted until the end of the simulation. Then we attached all spike trains to the input layer neurons in a one-to-one manner. The synaptic current triggered by the spike arriving at time $t_O$ is given by

$$I_{\text{syn}} = g_{\text{syn}}\left(v - E_{\text{syn}}\right) \tag{2}$$

$$g_{\text{syn}} = g_{\text{max}} e^{-(t-t_0)/\tau} \tag{3}$$

where $v$ is the post-synaptic voltage, the reversal potential $E_{\text{syn}} = 1$ mV, the maximum synaptic conductance $g_{\text{max}} = 0.05$ μS, and the time constant $\tau = 0.5$ ms.

Neurons in the input layer were modeled with a passive single-compartment model. The specific parameters were set as follows: membrane capacitance $c_{\text{m}} = 1.0$ μF cm$^{-2}$, membrane resistance $r_{\text{m}} = 10^4$ Ω cm$^2$, axial resistivity $r_{\text{a}} = 100$ Ω cm, reversal potential of passive compartment $E_{\text{l}} = 0$ mV.

The hidden layer contains a group of human pyramidal neuron models, receiving the somatic voltages of input layer neurons. The morphology was from Eyal, et al. [51], and all neurons were modeled with passive cables. The specific membrane capacitance $c_{\text{m}} = 1.5$ μF cm$^{-2}$, membrane resistance $r_{\text{m}} = 48,300$ Ω cm$^2$, axial resistivity $r_{\text{a}} = 261.97$ Ω cm, and the reversal potential of all passive cables $E_{\text{l}} = 0$ mV. Input neurons could make multiple connections to randomly-selected locations on the dendrites of hidden neurons. The synaptic current activated by the $k$-th synapse of the $i$-th input neuron on neuron $j$'s dendrite is defined as in Eq. (4), where $g_{ijk}$ is the synaptic conductance, $W_{ijk}$ is the synaptic weight, $f$ is the ReLU-like somatic activation function, and $v_i^t$ is the somatic voltage of the $i$-th input neuron at time $t$.

$$I_{ijk}^t = g_{ijk} W_{ijk} f(v_i^t) \tag{4}$$

$$f(v_i^t) = \begin{cases} v_i^t, & v_i^t > 0 \\ 0, & v_i^t \leq 0 \end{cases} \tag{5}$$

Neurons in the output layer were also modeled with a passive single-compartment model, and each hidden neuron only made one synaptic connection to each output neuron. All specific parameters

were set the same as those of the input neurons. Synaptic currents activated by hidden neurons are also in the form of Eq. (4).

## Image classification with HPC-Net

For each input image stimulus, we first normalized all pixel values to 0.0-1.0. Then we converted normalized pixels to spike trains and attached them to input neurons. Somatic voltages of the output neurons are used to compute the predicted probability of each class, as shown in equation 6, where $p_i$ is the probability of $i$-th class predicted by the HPC-Net, $\bar{v}_i$ is the average somatic voltage from 20 ms to 50 ms of the $i$-th output neuron, and $C$ indicates the number of classes, which equals the number of output neurons. The class with the maximum predicted probability is the final classification result. In this paper, we built the HPC-Net with 784 input neurons, 64 hidden neurons, and 10 output neurons.

$$p_i = \frac{\exp(\bar{v}_i)}{\sum_{c=0}^{C-1} \exp(\bar{v}_c)} \tag{6}$$

## Synaptic plasticity rules for HPC-Net

Inspired by previous work[36], we use a gradient-based learning rule to train our HPC-Net to perform the image classification task. The loss function we use here is cross-entropy, given in Eq. (7), where $p_i$ is the predicted probability for class $i$, $y_i$ indicates the actual class the stimulus image belongs to, $y_i = 1$ if input image belongs to class $i$, and $y_i = 0$ if not.

$$E = -\sum_{i=0}^{C-1} y_i \log p_i \tag{7}$$

When training HPC-Net, we compute the update for weight $W_{ijk}$ (the synaptic weight of the $k$-th synapse connecting neuron $i$ to neuron $j$) at each time step. After the simulation of each image stimulus, $W_{ijk}$ is updated as shown in Eq. (8):

$$W_{ijk} = W_{ijk} - \eta \frac{dt}{t_e - t_s} \sum_{t=t_s}^{t_e} \Delta W_{ijk}^t \tag{8}$$

$$\Delta W_{ijk}^t = \frac{\partial E}{\partial \bar{v}_j} r_{ijk} g_{ijk} f(v_i^t) \tag{9}$$

Here $\eta$ is the learning rate, $\Delta W_{ijk}^t$ is the update value at time $t$, $v_j$, $v_i$ are somatic voltages of neuron $i$ and $j$ respectively, $I_{ijk}$ is the $k$-th synaptic current activated by neuron $i$ on neuron $j$, $g_{ijk}$ its synaptic conductance, $r_{ijk}$ is the transfer resistance between the $k$-th connected compartment of neuron $i$ on neuron $j$'s dendrite to neuron $j$'s soma, $t_s = 30$ ms, $t_e = 50$ ms are start time and end time for learning respectively. For output neurons, the error term $\frac{\partial E}{\partial \bar{v}_j^{\text{out}}}$ can be computed as shown in Eq. (10). For hidden neurons, the error term $\frac{\partial E}{\partial \bar{v}_j^{\text{h}}}$ is calculated from the error terms in the output layer, given in Eq. (11).

$$\frac{\partial E}{\partial \bar{v}_j^{\text{out}}} = y_j - p_j \tag{10}$$

$$\frac{\partial E}{\partial \bar{v}_j^{\text{h}}} = \sum_{c=0}^{C-1} \frac{\partial E}{\partial \bar{v}_c^{\text{out}}} r_{jc} g_{jc} W_{jc} f'\left(\bar{v}_j^{\text{h}}\right) \tag{11}$$

Since all output neurons are single-compartment, $r_{jc}$ equals to the input resistance of the corresponding compartment, $r_c$. Transfer and input resistances are computed by NEURON.

Mini-batch training is a typical method in deep learning for achieving higher prediction accuracy and accelerating convergence. DeepDendrite also supports mini-batch training. When training HPC-Net with mini-batch size $N_{batch}$, we make $N_{batch}$ copies of HPC-Net. During training, each copy is fed with a different training sample from the batch. DeepDendrite first computes the weight update for each copy separately. After all copies in the current training batch are done, the average weight update is calculated and weights in all copies are updated by this same amount.

## Robustness against adversarial attack with HPC-Net

To demonstrate the robustness of HPC-Net, we tested its prediction accuracy on adversarial samples and compared it with an analogous ANN (one with the same 784-64-10 structure and ReLU activation, for fair comparison in our HPC-Net each input neuron only made one synaptic connection to each hidden neuron). We first trained HPC-Net and ANN with the original training set (original clean images). Then we added adversarial noise to the test set and measured their prediction accuracy on the noisy test set. We used the Foolbox[98,99] to generate adversarial noise with the FGSM method[93]. ANN was trained with PyTorch[100], and HPC-Net was trained with our DeepDendrite. For fairness, we generated adversarial noise on a significantly different network model, a 20-layer ResNet[101]. The noise level ranged from 0.02 to 0.2. We experimented on two typical datasets, MNIST[95] and Fashion-MNIST[96]. Results show that the prediction accuracy of HPC-Net is 19% and 16.72% higher than that of the analogous ANN, respectively.

## Reporting summary

Further information on research design is available in the Nature Portfolio Reporting Summary linked to this article.

## Data availability

The data that support the findings of this study are available within the paper, Supplementary Information and Source Data files provided with this paper. The source code and data that used to reproduce the results in Figs. 3–6 are available at https://github.com/pkuzyc/DeepDendrite. The MNIST dataset is publicly available at http://yann.lecun.com/exdb/mnist. The Fashion-MNIST dataset is publicly available at https://github.com/zalandoresearch/fashion-mnist. Source data are provided with this paper.

## Code availability

The source code of DeepDendrite as well as the models and code used to reproduce Figs. 3–6 in this study are available at https://github.com/pkuzyc/DeepDendrite.

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

## Acknowledgements

The authors sincerely thank Dr. Rita Zhang, Daochen Shi and members at NVIDIA for the valuable technical support of GPU computing. This work was supported by the National Key R&D Program of China (No. 2020AAA0130400) to K.D. and T.H., National Natural Science Foundation of China (No. 61088102) to T.H., National Key R&D Program of China (No. 2022ZD01163005) to L.M., Key Area R&D Program of Guangdong Province (No. 2018B030338001) to T.H., National Natural Science Foundation of China (No. 61825101) to Y.T., Swedish Research Council (VR-M-2020-01652), Swedish e-Science Research Centre (SeRC), EU/ Horizon 2020 No. 945539 (HBP SGA3), and KTH Digital Futures to J.H.K., J.H., and A.K., Swedish Research Council (VR-M-2021-01995) and EU/ Horizon 2020 no. 945539 (HBP SGA3) to S.G. and A.K. Part of the simulations were enabled by resources provided by the Swedish National Infrastructure for Computing (SNIC) at PDC KTH partially funded by the Swedish Research Council through grant agreement no. 2018-05973.

## Author contributions

K.D. conceptualized the project. K.D. and T.H. jointly supervised the project. Y.Z. and G.H. implemented DeepDendrite framework, conducted all experiments and performed data analysis. L.M. provided the support for high performance computing. Y.Z. and X.L. provided theoretical proof for DHS method. Y.Z., G.H. and K.D. wrote the draft of the manuscript. J.J.J.H., A.K., Y.H., S.Z., J.H.K., Y.T. and S.G. participated in discussions regarding the results. All authors contributed to the revision of the manuscript.

## Competing interests

The authors declare no competing interests.
