## [Peer Review File · Nature Communications]

A GPU-based computational framework that bridges Neuron simulation and Artificial IntelligenceREVIEWER COMMENTS

Reviewer #1 (Remarks to the Author):

The manuscript by Zhang et al. introduces a new numerical technique to speed up compartmental modeling of biologically detailed neurons. The authors develop a novel Dendritic Hierarchical Scheduling (DHS) method to accelerate the computation of compartmental models markedly. The authors apply this method to various compartmental models from the literature. They show how to run DHS on GPUs, thus speeding up the computation severalfold over the standard NEURON implementation. Finally, they apply this technique inside an artificial neural network, demonstrating that it may be possible to run biologically detailed ANNs within a reasonable time frame. While this is an essential direction of study, several problems may dampen this paper's possible impact.

Major

A possible problem with DHS is the morphology segmentation prior to the simulation's onset. This may drastically affect the computation, especially when active dendritic conductances are involved. As the authors are well aware, the electrotonic length of the dendritic and axonal branches is not fixed. Voltage-gated ion channels can expand or contract the effective size of dendritic compartments. Activation of sodium and calcium conductances decreases the effective size of the compartment, while activation of potassium conductance has the opposite effect. Therefore, segmenting the morphology before the onset of computation may be less efficient when dendritic conductances are activated. The authors display only somatic responses. It is thus hard to verify that DHS is accurate for dendritic regenerative activity. It may be that a revision of the algorithm can increase performance.

Disregarding the previous comment, the authors present a reasonably compelling description of DHS and its abilities in the first part of their paper. Then they apply the tool to simulate a neuron with realistic spine distribution and construct an ANN with a layer of compartmental models. These simulations are anecdotal, primarily demonstrating a possibility rather than providing a solution to a specific scientific problem. The technique's power is demonstrated in the first part of the manuscript, making it a reasonable methods paper. The second part does not add substantial scientific value. I think the authors should focus on the first part of the paper, only making this a clear methods paper.

For example, the authors use several models from the literature (Fig. 4). Several of these models contain excitable dendrites. However, they show only somatic traces in figure 4. Synaptic clustering should (especially in pyramidal neurons) generate dendritic calcium spikes. This can also be generated by dendritic current injection (another possible control for the accuracy of their method).

Using neurons with active dendrites for these simulations, the authors remove all active conductances from the dendrites when constructing their ANN (Fig. 6).

Why? Neurons with passive dendrites are not that different from the point neurons in the other layers of the ANN. Here the authors have the opportunity to insert a layer of neurons with active dendrites to test the effect on the network. They also tested only one type of neuron. Why not compare how an ANN with pyramidal neurons compares to one with mitral or Purkinje neurons.

In conclusion, this is an interesting methods paper to which a less rigorous part was added. Either remove this part and focus only on the method or try to expand it to demonstrate a straightforward solution to a scientific problem.

Minor

Figure 1: panels A-D are trivial and only repeat the Hines method.

Figure 1: panels F, H – the computational cost is ill-defined in the legend. Please add to the axis the notation #of operations.

Figure 2: panel F - here, the computational cost is noted as % on the axis, but the scale is between 0-1, suggesting this is just normalized. It also makes it problematic to compare to figure 1. Please use a similar axis on all figures.

Reviewer #2 (Remarks to the Author):

In this manuscript, Zhang et al. present a novel method, termed Dendritic Hierarchical Scheduling (DHS), to efficiently simulate compartmental neuronal models using GPUs. By modifying the CoreNEURON codes, they show significant speed-up in the simulations. The authors simulate morphologically detailed neuronal models with spines -simulated as two-compartment models. Spines are not typically simulated explicitly, as done here, due to the high computational cost associated with such detailed compartmental modelling. Interestingly, the authors show that simulating spine morphology is vital, as the traditional approach of increasing the specific capacitance of spiny compartments to account for the expanded area leads to artefacts. The authors also use their method to train an HPC-Net, consisting of morphologically detailed neuronal models –albeit with passive dendrites- on a machine learning task (image classification). They demonstrate robust performance, including against adversarial attacks, and show that this detailed network outperforms a traditional ANN consisting of point neurons.

The article is well written, and the presentation of the results is clear and concise. The methods section is detailed and self-explanatory. In my opinion, this manuscript is highly appropriate for Nature Communications as the proposed method, pending community validation, can be a game changer in the neuronal modelling field. While I am very positive about publishing this work in Nature Communications, I have some comments/suggestions that can help increase the robustness and impact of the proposed

methodology. My most important concern is the use of passive dendritic compartments that not only limits the flexibility and computational power of the implemented networks but is of significantly lower computational complexity. The authors should also demonstrate the computational benefits from using DHS when keeping track of ionic concentrations and/or when dendritic spikes are generated.

Major comments (not in order of importance nor appearance in the text)

1. Line 150: The authors build upon the observation that “computing ionic currents of each compartment is a fully independent process”. However, this independence is often subject to the channels used and the level of detail in a given biophysical model. For example, Ca^{2+} -dependent channels are not independent of other compartments as the Ca^{2+} concentration and its diffusion along the longitudinal axis is affected by the activity in multiple compartments and this in turn affects the activation/kinetics of these conductances. In such a case, where multiple compartments contribute to the activities of specific ion channels, I expect that the splitting level will decrease? The authors should provide more information about the ionic mechanisms present in the compared models and whether they used the rxd package of NEURON (e.g. to account for diffusion processes).
2. The authors explicitly state that their proposed method is faster than CoreNeuron and traditional CPU-based methods without loss of accuracy. This is great but in addition to the quantitative comparisons, the authors should explain in more detail the methodological and conceptual differences between the CoreNeuron and the DHS approaches and -if relevant- highlight the conditions under which DHS is (or is expected to be) faster/better.
3. The claimed equivalence in terms of simulation accuracy is only supported by Figure S4 whereby a very simple experimental protocol is used. Accuracy in biophysical models does not only refer to somatic voltage responses or the generation of a single action potential. It also refers to synaptic currents, ionic currents, and the various ionic species. Thus, a more realistic biophysical model that incorporates active dendritic conductances (e.g., Na^+ , Ca^{2+} , K^+ channels) and a simulation protocol comparing multiple features (e.g., dendritic spiking, back-propagating spikes, bursting etc.) is needed in order to demonstrate that the same “accuracy” can be achieved with DHS versus the traditional Hines method. I would suggest demonstrating the equivalence of the two approaches when high-frequency stimuli drive a more complex biophysical model, resulting in dendritic spiking and bursting at the cell body.
4. Lines 353-354: The authors state that they use the same hyperparameters. How do they ensure that the ANN has the same activation function(s) as the biophysical models? For the biophysical models, one can assume that the activation function is the input-output curve (see Fig. 5D). In ANNs this is not clear. Did the authors use a sigmoid function centered at zero?
5. Lines 373-374: The authors state that dendrites are the key mediator of improved performance under adversarial attacks, which is a conclusion reached after comparing ANNs with their HPC-Net. But a clear explanation of this finding is missing and is rather unintuitive given that dendrites are modelled as passive cables (e.g., see Chavlis and Poirazi, 2021; Acharyaa et al, 2021). Which dendritic feature improves robustness to adversarial attacks? And how would the simulation of active dendritic mechanisms change these findings? I would suggest presenting at least one example/comparison using a biophysical model with active dendritic conductances.

6. Importantly, other studies have also shown the impact of dendrites in improving performance on computer vision tasks. For example, Jones and Kording, 2021; Iyer et al., 2022; Bird et al., 2021, etc. I suggest citing and discussing these studies in the manuscript, especially with respect to interpretability of their improved performance findings.

Minor comments (not in order of importance nor appearance in the text)

1. Line 73, Introduction: I suggest adding a verb before “bridge” or using the expression “bridge the gap”, i.e., “[...] were utilized to be the bridge between [...]” or “[...] were utilized to bridge the gap between [...]”.
2. Line 69-70: Please add “[...] such as ion channels, excitatory/inhibitory synaptic currents, etc. [...]” for clarity.
3. Line 79: How do you define “general intelligence”? I suggest adding a sentence with some references in specific domains where AI performs poorly on multiple tasks or exhibits catastrophic forgetting.
4. Figure S1: I suggest adding the y-axis label “Computational Cost” in both the top and bottom left panels and the x-axis label in the middle of the top row panels.
5. Lines 167, 227, 252, etc.: Please add a comma after “i.e.” or “e.g.” such that “i.e.,” or “e.g.,” OR remove the comma from such statements for consistency across the manuscript.
6. Line 174: Please write the “k” in “k parallel threads” with italic as in line 184 for consistency.
7. Line 193: Please cite the ModelDB as: “McDougal RA, Morse TM, Carnevale T, Marengo L, Wang R, Migliore M, Miller PL, Shepherd GM, Hines ML. Twenty years of ModelDB and beyond: building essential modeling tools for the future of neuroscience. *Journal of Computational Neuroscience* 2017; 42(1):1-10 (2017).”. See here for more details: [https://senselab.med.yale.edu/modeldb/HowToCite#:~:text=When you use ModelDB to,The ModelDB \(McDougal et al.](https://senselab.med.yale.edu/modeldb/HowToCite#:~:text=When you use ModelDB to,The ModelDB (McDougal et al.)
8. Line 307: The field of “deep learning” consists of more architectures than DNNs. I suggest rephrasing the sentence.
9. Lines 312-321: Please add respective references for the “recent work” (line 312) and the “foundational work” (line 315) mentioned in the text.
10. Line 451: Please add “[...] increase the internal [Ca²⁺]”. The superscript “2+” should be placed inside the squared brackets.
11. Line 509: Please add some indicative references for the “recent studies” mentioned here.

Reviewer #3 (Remarks to the Author):

I find the results of the adversarial attacks promising (Figure 6e), but feel that more analysis would need to be done in order to be convinced that dendritic computations confer such robustness (otherwise, I think such claims should be toned down in the manuscript).

This ultimately boils down to recommending the following:

1. More challenging, ecologically plausible visual task (e.g. ImageNet)
2. Dendrites implanted into deeper networks (that are known to match primate visual cortex, e.g. ResNet-18)
3. More networks through which the adversarial examples are generated from
4. More methods of generating adversarial attacks beyond FGSM (which are easily testable since you used the Bethge Lab Foolbox library)

1-3 are important to demonstrate at a minimum, and having 4 on ImageNet would **really** strengthen the result.

Specifically, the reasoning for points 1 & 2 come from the following observations:

In lines 351-352, the base network of comparison is too simple, given that the MNIST performance is low (good networks get at least 99% on this task). With that desired goal of 99% accuracy in mind, it is also clear that this is too simple of a task to separate models, especially when making claims of biological plausibility. This is evident, for example, with biologically plausible learning rules, where training on MNIST with feedback alignment leads to similar performance as backpropagation, but is clearly a suboptimal choice on ImageNet (Bartunov et al. NeurIPS 2018). Furthermore, lots of work has shown that training ANNs on ImageNet has yielded neurally plausible representations (Yamins et al. 2014, Hong et al 2016; Khaligh-Razavi and Kriegeskorte 2014, Guclu and van Gerven 2015, and others), underscoring the importance of training on this imageset when making claims of biological plausibility conferring a behavioral benefit (in this case, increased adversarial robustness).

Not only that, but training on ImageNet should be feasible since you used A100s, and would further underscore the first part of the paper which deals with the efficiency of DeepDendrite. My specific suggestion would be to have ResNet-18 as your base ANN model of comparison, and implant your DeepDendrite into it. I would be curious especially if it could match the performance of deeper models, such as ResNet-50, since dendritic computations tend to have lots of additional nonlinear operations. If that was true, that would **further** strengthen the result (as an additional behavioral metric beyond

adversarial robustness), and if not that is fine too, so long as adversarial robustness is maintained still on this imageset.

The reasoning for points 3 & 4 come from the following observations:

In lines 369-374, it is great that you are using a transfer attack, since this is generally one of the harder cases of adversarial attacks. To really show this result is robust, I would choose multiple base models to generate adversarial examples from (on ImageNet), as well as additional methods of generating these attacks (which seems reasonably easy to try given you are using Foolbox's implementations).

We would like to thank reviewers for his/her careful review of our manuscript, which have been critical in improving the quality and impact of our work. We are particularly delighted that Reviewer 1 sees our work as presenting “an essential direction”, Reviewer 2 recognizes its potential “to be a game changer” and Reviewer 3 finds our results to be “promising”. We have carefully considered each of your suggestions and made point-to-point revisions to address all concerns.

Below, we outline the major changes we have made to our manuscript in response to your comments:

1. We have made significant updates to our *DeepDendrite* framework, which includes substantial algorithmic and technical improvements that enable the support of larger and deeper biophysically detailed dendritic neural networks (BDNNs).
2. We have validated the numerical accuracy of our DHS method on neuron models with active dendrites and different segmentation strategies, as suggested by the reviewers (Supplementary Fig. 7).
3. We have investigated the role of passive dendrites in resisting adversarial attacks, in order to address reviewer's concerns regarding the mechanisms underlying our AI task (Supplementary Fig. 9).
4. We have constructed more complex and deeper BDNNs and applied them to perform more challenging tasks, in response to reviewer's concerns raised about the results of the adversarial attacks (Supplementary Figs. 10 and 11).
5. We have validated the performance of our models under different base models (ResNet, VGG) and adversarial attack methods (FGSM, PGD, and DeepFool) in more challenging tasks to address reviewer's concerns about the validity of our adversarial attack task (Supplementary Figs. 10 and 11).

We believe that these revisions have significantly strengthened the scientific validity and robustness of our work. We hope that our response has adequately addressed your concerns and that you will find our revised manuscript suitable for publication in *Nature Communications*.

Reviewer #1 (Remarks to the Author):

“The manuscript by Zhang et al. introduces a new numerical technique to speed up compartmental modeling of biologically detailed neurons. The authors develop a novel Dendritic Hierarchical Scheduling (DHS) method to accelerate the computation of compartmental models markedly. The authors apply this method to various compartmental models from the literature. They show how to

run DHS on GPUs, thus speeding up the computation severalfold over the standard NEURON implementation. Finally, they apply this technique inside an artificial neural network, demonstrating that it may be possible to run biologically detailed ANNs within a reasonable time frame. While this is an essential direction of study, several problems may dampen this paper's possible impact."

We appreciate your recognition of the significance of our study and the direction of our research.

Major

"A possible problem with DHS is the morphology segmentation prior to the simulation's onset. This may drastically affect the computation, especially when active dendritic conductances are involved. As the authors are well aware, the electrotonic length of the dendritic and axonal branches is not fixed. Voltage-gated ion channels can expand or contract the effective size of dendritic compartments. Activation of sodium and calcium conductances decreases the effective size of the compartment, while activation of potassium conductance has the opposite effect. Therefore, segmenting the morphology before the onset of computation may be less efficient when dendritic conductances are activated. The authors display only somatic responses. It is thus hard to verify that DHS is accurate for dendritic regenerative activity. It may be that a revision of the algorithm can increase performance."

Response:

We thank the reviewer for raising the important question regarding the impact of the morphology segmentation process on the accuracy of the Dendritic Hierarchical Scheduling (DHS) method, particularly in the presence of active conductance.

Motivations

It is essential to clarify that the morphology segmentation is performed by the simulator, in accordance with the strategy specified by the user, prior to the initiation of the simulation process. This segmentation process is separate from the DHS method, and the results of the segmentation are fixed throughout the simulation. The DHS method is then applied to the already segmented neuron models before the simulation starts.

We agree with the reviewer that the electrotonic length of the dendritic and axonal branches is impacted by the presence of active conductances. Our common experience with the NEURON simulator supports the notion that the responses of a detailed model segmented into 100 compartments may differ from those of the same model segmented into 500 compartments. However, it is the user's responsibility to choose an appropriate segmentation strategy that aligns with their intended tasks. The DHS method operates on

the fixed segmentation strategy and does not alter it in any way. Therefore, a fair comparison of the accuracy of the DHS method to the original Hines method can only be made in a segmented model that has already been defined by the user.

The DHS method has been mathematically proven to maintain the accuracy of the simulation results in the presence of dendritic active conductances. As we stated in the manuscript, the process of “computing ionic currents and solving linear equations are two critical phases” in the simulation of biologically detailed neurons. The computation of ionic currents of each compartment is an independent process, which can be parallelized on devices. The DHS method is designed specifically to address the linear equation solving phase and ensure its accuracy.

Our mathematical proof demonstrates that the DHS method is based on parallel computing theory and can maintain the accuracy of the simulation results, comparable to the original Hines Method, regardless of the chosen segmentation strategy. This is important as it indicates that the DHS method is robust to changes in the segmentation strategy, as long as the strategy is fixed prior to the onset of the simulation.

Experiments & Results

To further validate the numerical accuracy of our DHS method, we conducted additional experiments using the layer 5 pyramidal neuron model with active dendrites (Shai, Anastassiou et al. 2015) and different segmentation strategies (Supplementary Fig. 7), and compared the results with those obtained using the NEURON simulator. Our experiments involved both synaptic inputs on dendrites and current injections on the soma. Both AMPA-based and NMDA-based synaptic currents were simulated, and the voltage responses were recorded from the dendrites to the soma. Our results show that the recorded dendritic events and somatic responses from our DeepDendrite framework are identical to those obtained using the NEURON simulator.

Conclusions

We have clarified that the DHS method operates on a fixed segmentation strategy chosen by the user and maintains its accuracy, as demonstrated by mathematical proof and additional experiments using the layer 5 pyramidal neuron model with active dendrites and different segmentation strategies. The results show that our DeepDendrite framework produces dendritic and somatic voltage responses identical to those obtained using the NEURON simulator.

“Disregarding the previous comment, the authors present a reasonably compelling description of DHS and its abilities in the first part of their paper. Then they apply the tool to simulate a neuron

with realistic spine distribution and construct an ANN with a layer of compartmental models. These simulations are anecdotal, primarily demonstrating a possibility rather than providing a solution to a specific scientific problem. The technique's power is demonstrated in the first part of the manuscript, making it a reasonable methods paper. The second part does not add substantial scientific value. I think the authors should focus on the first part of the paper, only making this a clear methods paper.“

Response:

Thank you for the thoughtful suggestions. We appreciate your positive comments on the first part of our paper and agree that it provides a compelling description of DHS and its abilities. We also understand your concerns on the second part of our paper, where we apply DHS to simulate a neuron with realistic spine distribution and construct a neural network with a layer of morphologically detailed models. However, we believe that the demonstrations showcase the practical application of DHS and its potential in modeling complex biological systems, thereby advancing our understanding of brain function and the development of brain-inspired artificial intelligence (AI). These demonstrations serve as a crucial proof-of-concept for the potential of DHS in establishing brain-like AI systems, which can have significant implications for future scientific investigations.

While we appreciate your suggestion to focus solely on the first part of the paper as a methods paper, we believe that presenting the full extent of our work provides a more comprehensive understanding of the capabilities of DHS. We will, however, take your suggestions into consideration by revising the text and adding further experiments to strengthen our work.

Revisions

we have added the following statement at the end of the section “*DHS Creates Cell-type-specific Optimal Partitioning*” to clarify our intention:

“It is important to emphasize that the primary goal of employing biophysically detailed dendritic neural networks in AI tasks is not to replicate the success of ANNs. Instead, it aims to provide a proof-of-concept demonstration of the potential advantages of dendritic learning in AI by drawing upon well-established ANN theory.”

“For example, the authors use several models from the literature (Fig. 4). Several of these models contain excitable dendrites. However, they show only somatic traces in figure 4. Synaptic clustering should (especially in pyramidal neurons) generate dendritic calcium spikes. This can also be generated by dendritic current injection (another possible control for the accuracy of their method).”

Response:

Experiments & Results

Thank you for the suggestions! As per your suggestion, we used the same detailed model (Shai, Anastassiou et al. 2015) as in the previous segmentation experiments, which includes sodium, potassium, and calcium channels on both dendrites and the soma (Supplementary Fig. 7). To trigger synaptic currents, we generated 400 Poisson trains at a frequency of 2.5 Hz, applied from 80 ms to 400 ms, which included both AMPA and NMDA currents. In addition, we injected a 0.5 nA current to the soma, which was sustained for 100 ms. Instead of using dendritic current injections as suggested by the reviewer, we chose to use a large number of random synaptic inputs because it is commonly believed that massive random synaptic events are more challenging to test a model's numerical accuracy.

Our results indicate that the dendritic and somatic responses simulated using DeepDendrite were identical to those generated using NEURON (Supplementary Fig. 7). This suggests that our tool can accurately simulate the dynamics of a neuron with excitable dendrites, including the generation of dendritic spikes (Supplementary Fig. 7). From the experimental results we can see that the massive synaptic inputs triggered dendritic spikes. When the current stimulus is injected to the soma, it induced spike bursting, and the somatic spikes then backpropagate to dendrites (Supplementary Fig. 7). Our DeepDendrite framework can reproduce all these complicated dendritic and somatic events, and get identical results to NEURON.

Revisions

We have added the contents about numerical accuracy at the end of the section "*Speeding Up DHS by GPU Memory Boosting*":

"Moreover, compared to the conventional serial Hines method in NEURON running with a single-thread of CPU, DHS speeds up the simulation by 2-3 orders of magnitude (Supplementary Fig. 3), while retaining the identical numerical accuracy in the presence of dense spines (Supplementary Figs. 4 and 8), active dendrites (Supplementary Fig. 7) or different segmentation strategies (Supplementary Fig. 7)."

"Using neurons with active dendrites for these simulations, the authors remove all active conductances from the dendrites when constructing their ANN (Fig. 6).

Why? Neurons with passive dendrites are not that different from the point neurons in the other layers of the ANN. Here the authors have the opportunity to insert a layer of neurons with active dendrites to test the effect on the network. They also tested only one type of neuron. Why not compare how an ANN with pyramidal neurons compares to one with mitral or Purkinje neurons.

In conclusion, this is an interesting methods paper to which a less rigorous part was added. Either remove this part and focus only on the method or try to expand it to demonstrate a straightforward solution to a scientific problem.”

Response:

We thank the reviewer for raising critical concerns regarding our use of passive dendrites in the proposed neural network model (i.e., BDNN). We would like to reiterate that the primary focus of our work is centered on the development and application of the DHS method and DeepDendrite framework. The neural networks with morphologically detailed neurons serve as a proof-of-concept to illustrate the potential of our proposed methodology and framework in advancing current state-of-the-art (SOTA) brain-like AI systems. Below, we will provide further details to explain our motivations and perform additional experiments to address the reviewer's concerns.

Rationale and hypotheses for using passive dendrites in BDNNs

we decided to employ only passive dendrites due to the absence of a suitable learning method for active dendrites to perform image classification tasks. Despite recent advancements in dendritic neuron models and their application to AI tasks, most SOTA approaches rely on simplified dendritic models that only contain 2-3 compartments of passive dendrites (Guerguiev, Lillicrap et al. 2017, Sacramento, Costa et al. 2018, Payeur, Guerguiev et al. 2021). Additionally, the learning methods proposed for morphologically detailed neurons with active mechanisms have been primarily limited to single neurons and have yet to be applied to large-scale networks in relatively challenging tasks, e.g., classical image classification (Bicknell and Häusser 2021, Moldwin, Kalmenson et al. 2021).

It's worth noting that the overall performance of the proposed BDNN models is a combined effect of *architecture design, learning algorithms, training techniques, and active/passive dendritic mechanisms*. There remain numerous challenges, such as how to better combine training techniques (e.g., *hyperparameter tuning, input preprocessing, choice of the optimizer*, etc.) with active or passive dendritic mechanisms. Therefore, extending the learning method from single neurons to large-scale neural networks with active dendrites requires additional significant algorithmic innovations and technical improvements in the training framework, which falls outside the scope of our current work.

We acknowledge the reviewer's suggestion to investigate the effects of active dendrites on the network, and we appreciate the value of exploring this avenue. However, we firmly believe that our utilization of morphologically detailed neural networks with passive dendrites represents a substantial step forward compared to current dendritic ANNs that rely on simplified passive dendrites.

To gain further insights into the role of passive dendrites in BDNNs, we conducted additional experiments with elaborated synaptic connectivity to investigate their function in image classification tasks.

We hypothesize that the spatiotemporal filtering feature of passive dendrites improves the robustness of BDNNs to adversarial attacks. In the experiment settings, we converted input images to spike trains and attached them to the neurons, then added adversarial noise to “perturb” the spike timing. As dendrites are typically modeled as passive cables, Cable theory predicts that their ability to filter dendritic responses during propagation to the soma renders them less susceptible to the “perturbations” in spike timing induced by (adversarial) noise (London and Häusser 2005, Poirazi and Papoutsis 2020). Specifically, our key learning algorithms (Eq. 8 And Eq.9 in the main manuscript, as below) incorporate this concept:

$$W_{ijk} = W_{ijk} - \eta * \frac{dt}{t_e - t_s} \sum_{t=t_s}^{t_e} \Delta W_{ijk}^t$$

$$\Delta W_{ijk}^t = \frac{\partial Loss}{\partial \bar{v}_j} * r_{ijk} * g_{ijk} * f(v_i^t)$$

Here, r_{ijk} is the transfer impedance between the neuron j 's dendrite to the soma computed by NEURON, which may contribute to the filtering of “perturbations”. This feature is a unique aspect in our learning algorithms due to the inclusion of passive dendrites (cables).

Based on this hypothesis, we further predict that placing synaptic inputs located more distally will increase the BDNN's ability to resist adversarial attacks.

Experiments & Results

To test this hypothesis, we compared the robustness of two BDNNs with different connectivity patterns: one with the input neurons connected to the proximal dendrites (< 100 μm) of the hidden neurons, and the other with the input neurons connected to the distal dendrites (> 200 μm) of the hidden neurons (Supplementary Fig. 9a). We also extended the number of training epochs to 60. Apart from these, other model settings as well as training and evaluating procedures are the same as Fig. 6d in the manuscript. The classification accuracy for the two BDNNs as well as the corresponding ANN at different FGSM adversarial noise levels is as in Supplementary Fig. 9b. The BDNNs reached almost the same accuracy as ANN on clean examples (attack strength $\epsilon=0$: 97.23% for proximal, 97.25% for distal, and 97.65% for ANN). As attack strength increased, the accuracy for

ANN soon began to drop, while for BDNNs the accuracy almost remained held. Under large attack strength, the accuracy for proximally- and distally-connected networks began to diverge, with 88.65% and 92.96% at attack strength $\varepsilon=0.2$ respectively, while still outperforming ANN (78.40%) by a large margin. Our results indicated that the distally-connected model is more robust against large adversarial noise than the proximally-connected model, suggesting that passive dendrites, together with structured synaptic connectivity, can potentially make dendritic neural networks more robust in noisy environments.

More efforts are required in the future to further optimize the design of large-scale dendritic neural networks and exploit their potential on more challenging tasks, such as ImageNet classification. We believe the DeepDendrite framework is able to provide a valuable computational tool for making such explorations possible.

Discussions on exploring more neuronal types

Finally, with regard to the reviewer's suggestion to test more neuronal types in the BDNN models, we agree that this is a very interesting proposal and represents another essential direction for exploring possible dendritic mechanisms in these models. While we could not address this suggestion in the current manuscript due to content limitations, we plan to systematically investigate this topic in the future, covering nearly all principal neurons in the brain. The DeepDendrite framework will provide a solid foundation to make this ambition possible.

Conclusions

We have emphasized that our work focuses on the development and application of the DHS method and DeepDendrite framework, using neural networks with morphologically detailed neurons as a proof-of-concept. We chose passive dendrites due to the lack of suitable learning methods for active dendrites in image classification tasks. Additional experiments were conducted to investigate the role of passive dendrites in BDNNs, revealing that structured synaptic connectivity and passive dendrites can enhance the network's robustness in noisy environments. The DeepDendrite framework will provide a valuable computational tool for future exploration and optimization of large-scale dendritic neural networks.

Revisions

we have added the additional experiment and results in the Discussion part, "In addition, we investigated the roles passive dendrites play in resisting adversarial attacks. We hypothesized that passive dendrites could serve as spatiotemporal filters, with distal dendrites being more robust to noise interference than proximal dendrites when signals

are propagating along the dendrite. To investigate this, we compared two HPC-Nets with distinct connectivity patterns: one with input neurons connected to the proximal part of basal dendrites ($<100\ \mu\text{m}$) and another to the distal part ($>200\ \mu\text{m}$) (Supplementary Fig. 9a). Our results revealed that both HPC-Nets demonstrated greater robustness against adversarial noise than the regular ANN, with the distally-connected model (92.96% accuracy, attack strength $\epsilon=0.2$) outperforming the proximally-connected model (88.65% accuracy, $\epsilon=0.2$) (Supplementary Fig. 9b). This finding suggests passive dendrites may indeed play a role in defending against adversarial attacks by acting as spatiotemporal filters.”

Minor

Figure 1: panels A-D are trivial and only repeat the Hines method.

Response:

We thank the reviewer to raise the concern regarding the content of Figure 1. Although panels A-D seem to be a mere repetition of the Hines method, we believe they serve a crucial purpose in illustrating the context and background for our proposed DHS method.

During our conference and seminar presentations, we have frequently encountered questions regarding the specifics of the Hines method. We then realize that despite it being considered as a “gold standard” for solving cable equations, the implementation details of the method are not widely known for many in the field. As a result, we have included panels A-D to provide a clear and concise overview of the Hines method for our readers.

By doing so, we aim to ensure that readers have a solid understanding of the Hines method, thereby facilitating a more comprehensive evaluation of our DHS method. We believe that this approach enhances the overall clarity and accessibility of our work for diverse readers. Therefore, we believe they are essential for providing the necessary background and establishing the motivation for our proposed DHS method.

Figure 1: panels F, H – the computational cost is ill-defined in the legend. Please add to the axis the notation #of operations.

Response:

Thank you for your suggestion, we agree it’s important to reiterate our formulation of computational cost (introduced in the first paragraph of the section, “*Dendritic Hierarchical Scheduling (DHS) Method*” in the Results part) for clarity. We’ve added to the axis the notation “# of steps”. We’ve also added “steps taken in the equation solving phase” to the notes for figure 1f as well.

Figure 2: panel F - here, the computational cost is noted as % on the axis, but the scale is between 0-1, suggesting this is just normalized. It also makes it problematic to compare to figure 1. Please use a similar axis on all figures.

Response:

Regarding the "%" symbol: You are correct that the "%" notation is not appropriate in this context. We have removed the "%" and updated the axis notation to "Relative Cost" in Figure 2F, along with the corresponding notes for the figure.

As for using a similar axis on all figures: We appreciate the reviewer's concern, but we believe that using absolute computational cost in Figure 1 and relative cost in Figure 2 serves distinct purposes for each figure. In Figure 1, panels F and H show the absolute computational cost for the Hines method, which corresponds to the number of matrix rows/columns in panel E (i.e., 838 for the pyramidal model and 36,218 for the pyramidal model with spines). We believe that using absolute cost values in Figure 1 allows our readers to gain a clearer understanding when examining panels E through H.

Conversely, in Figure 2F, our primary goal is to demonstrate the significant reduction in computational cost when implementing the DHS method across all tested neuron models. Since the absolute cost of the serial Hines method varies by orders of magnitude for different models (as shown in Figure 1F), using relative cost values in Figure 2 allows for a clearer comparison of DHS performance against the serial Hines method across all models. Therefore, we believe it is justified to use different axes in Figures 1 and 2.

Reviewer #2 (Remarks to the Author):

“In this manuscript, Zhang et al. present a novel method, termed Dendritic Hierarchical Scheduling (DHS), to efficiently simulate compartmental neuronal models using GPUs. By modifying the CoreNEURON codes, they show significant speed-up in the simulations. The authors simulate morphologically detailed neuronal models with spines -simulated as two-compartment models. Spines are not typically simulated explicitly, as done here, due to the high computational cost associated with such detailed compartmental modelling. Interestingly, the authors show that simulating spine morphology is vital, as the traditional approach of increasing the specific capacitance of spiny compartments to account for the expanded area leads to artefacts. The authors

also use their method to train an HPC-Net, consisting of morphologically detailed neuronal models –albeit with passive dendrites- on a machine learning task (image classification). They demonstrate robust performance, including against adversarial attacks, and show that this detailed network outperforms a traditional ANN consisting of point neurons.

The article is well written, and the presentation of the results is clear and concise. The methods section is detailed and self-explanatory. In my opinion, this manuscript is highly appropriate for Nature Communications as the proposed method, pending community validation, can be a game changer in the neuronal modelling field. While I am very positive about publishing this work in Nature Communications, I have some comments/suggestions that can help increase the robustness and impact of the proposed methodology. My most important concern is the use of passive dendritic compartments that not only limits the flexibility and computational power of the implemented networks but is of significantly lower computational complexity. The authors should also demonstrate the computational benefits from using DHS when keeping track of ionic concentrations and/or when dendritic spikes are generated.”

We sincerely thank the reviewer for acknowledging the significance of our study and recognizing that it has the potential to be a game changer in the field of neuronal modeling.

“Major comments (not in order of importance nor appearance in the text)

1. Line 150: The authors build upon the observation that “computing ionic currents of each compartment is a fully independent process”. However, this independence is often subject to the channels used and the level of detail in a given biophysical model. For example, Ca²⁺-dependent channels are not independent of other compartments as the Ca²⁺ concentration and its diffusion along the longitudinal axis is affected by the activity in multiple compartments and this in turn affects the activation/kinetics of these conductances. In such a case, where multiple compartments contribute to the activities of specific ion channels, I expect that the splitting level will decrease? The authors should provide more information about the ionic mechanisms present in the compared models and whether they used the rxd package of NEURON (e.g. to account for diffusion processes).”

Response:

We thank the reviewer for raising a critical concern regarding the parallelization of computing ionic mechanisms in our method. First, we would like to provide further clarification on the statement “computing ionic currents of each compartment is a fully independent process”, which is derived from the original CoreNEURON paper detailing the parallel computing implementation for ionic mechanisms. As the DHS method is built upon CoreNEURON, it naturally inherits this feature. In the following response, we will first discuss the “set up equations” process within CoreNEURON, and then explain how the

calculation of ionic concentrations and their diffusion within this phase does not adversely affect the DHS method.

Discussions on “set up equations” process

The CoreNEURON simulator consists of two main processes: *computing ionic currents* and *solving linear equations*. The simulator first calculates ionic currents for each compartment, which are then used to formulate the linear equations. Subsequently, the simulator solves the equations to determine voltage increments. For further technical details, figure 1b depicts the entire simulation phase as four sub-processes: 1) initialize; 2) set up equations; 3) solve linear equations; and 4) update states.

The computation of ionic currents takes place within the second sub-process, “set up equations”. At this stage, the simulator calculates the ionic and synaptic currents for all compartments in parallel, at the compartment level. The results are then integrated into the Hines matrix and right-hand-side values to generate the equations. After computing ionic and synaptic currents for each compartment, the simulator calculates the axial currents between neighboring compartments, also in parallel at the compartment level, and incorporates these results into the matrix to generate the equations.

Discussions on the calculation of ionic concentrations

Regarding the calculation of ionic concentrations and their diffusion, it should be emphasized that these calculations require only a minimal number of operations during the “set up equations” phase. The original CoreNEURON simulator stores ion concentrations using additional variables. Throughout the simulation, different mechanisms of the same ion type within various compartments modify the concentration values based on their respective ion currents. These changes in concentration influence the reversal potential and subsequently affect the ion currents in future time steps. The diffusion between compartments is also incorporated in the equations (with non-diagonal matrix values representing coefficients between compartments), so the interactions among all compartments are further considered once the equations are solved.

The pseudo-code below demonstrates how the simulator calculates ionic currents and ion concentrations within the “set up equations” phase. Note that updating ionic concentration only requires a single “atomic operation” in GPU (step 4 in pseudo-code), which is an efficient and conflict-free operation to perform concurrent updates to shared memory locations without race conditions or conflicts, ensuring the correct execution of parallel processes.

Algorithm SetupMatrix

Input: Voltages and Mechanisms on all compartments

Output: The values in the Hines matrix

- 1: **for** each mechanism in all mechanisms:
 - 2: parallelly compute the ionic currents generated by this mechanism on all its corresponding compartments
 - 3: **if** current mechanism is an ion channel
 - 4: **update** the corresponding ionic concentration with an *atomic operation* (the change of ionic concentration will affect the ionic currents in the next time step)
 - 5: **end if**
 - 6: **end for**
 - 7: parallelly compute the axial currents among neighbor compartments
-

The interactions among compartments are represented by the linear equations (with non-diagonal matrix values representing coefficients between compartments). Once these equations are solved, the influence of different compartments' ionic and synaptic currents is also considered. Due to the linear equations accounting for the influence between neighboring compartments and the parallel computation of ionic currents and concentrations using atomic operations in GPU, the DHS method remains unaffected.

Lastly, we would like to clarify that, in the current stage, we do not utilize the rxd package of NEURON. However, as long as CoreNEURON supports the rxd package, incorporating it into DeepDendrite should not pose a significant technical challenge.

Conclusions

In summary, we have addressed the concern about the independence of computing ionic mechanisms in our method. We have explained the processes within CoreNEURON and how the DHS method handles the calculation of ionic concentrations and their diffusion with atomic operations that avoid negatively impacting the parallelization. Additionally, we have clarified that the rxd package of NEURON is not currently utilized, but its incorporation should be feasible once supported by CoreNEURON.

“2. The authors explicitly state that their proposed method is faster than CoreNeuron and traditional CPU-based methods without loss of accuracy. This is great but in addition to the quantitative comparisons, the authors should explain in more detail the methodological and conceptual differences between the CoreNeuron and the DHS approaches and -if relevant- highlight the

conditions under which DHS is (or is expected to be) faster/better.”

Response:

We appreciate the reviewer's important questions regarding the differences between the CoreNEURON and DHS approaches. As DHS is built upon the CoreNEURON solver, it can be considered an optimized, finest-grained parallel method for CoreNEURON. The methodological and conceptual differences between DHS and CoreNEURON primarily lie in the parallelization strategies used for solving equations.

Discussions

CoreNEURON adopts the *cuHinesBatch* method (Valero-Lara et al. 2017), a network-level parallel method that distributes solving equations phases among different threads for each cell. The *cuHinesBatch* method has also been included in standard CUDA libraries and can be viewed as the state-of-the-art method for solving Hines method on GPU. However, it is important to note that CoreNEURON only parallelizes the solving equations phase for each cell using one thread, and does not parallelize the computation inside a single cell.

DHS goes further by parallelizing the computation inside each cell using an optimal compartment-grained parallelization strategy that fully utilizes multiple threads to perform the solving phase for individual cells. We integrate this compartment-grained parallelization strategy into the CoreNEURON solver, enabling the DHS method to benefit from both compartment-grained and network-level parallelization.

By fully utilizing the parallel computing capabilities of GPUs, DHS achieves higher speedup than CoreNEURON. Modern GPUs consist of massive computing units, such as CUDA cores in Tesla V100/A100, that can be effectively utilized by DHS to achieve higher computational efficiency. In general, when a neuron model contains complex dendrites with many compartments, DHS can parallelize more compartments to achieve greater speedup. DHS is particularly effective for neurons with highly elaborate dendritic patterns and dense spines, such as cortical/ hippocampal pyramidal neurons and cerebellar Purkinje neurons. Our experiments indicate that the DHS method can achieve ~10x speedup in this case. Conversely, when a neuron contains fewer dendritic branches, fewer compartments can be parallelized, which leads to lower speedup. Nonetheless, DHS is expected to be faster than the CoreNEURON solver on GPU in all cases.

Conclusions

We have clarified that the DHS approach extends the CoreNEURON solver by employing both compartment-grained and network-level parallelization strategies, fully utilizing the parallel computing capabilities of GPUs. DHS excels particularly in neuron models with

complex dendrites and many compartments, such as cortical/hippocampal pyramidal neurons and cerebellar Purkinje neurons. Overall, DHS is expected to be faster than the CoreNEURON solver on GPU in all cases, with experiments demonstrating up to ~10x speedup.

3. The claimed equivalence in terms of simulation accuracy is only supported by Figure S4 whereby a very simple experimental protocol is used. Accuracy in biophysical models does not only refer to somatic voltage responses or the generation of a single action potential. It also refers to synaptic currents, ionic currents, and the various ionic species. Thus, a more realistic biophysical model that incorporates active dendritic conductances (e.g., Na⁺, Ca²⁺, K⁺ channels) and a simulation protocol comparing multiple features (e.g., dendritic spiking, back-propagating spikes, bursting etc.) is needed in order to demonstrate that the same “accuracy” can be achieved with DHS versus the traditional Hines method. I would suggest demonstrating the equivalence of the two approaches when high-frequency stimuli drive a more complex biophysical model, resulting in dendritic spiking and bursting at the cell body.

Response:

Experiments

We are grateful for the valuable suggestions raised by the reviewer. In accordance with your recommendation, we utilized the DeepDendrite simulation tool to model the layer 5 pyramidal neuron system proposed by *Shai et al. (2015)*(Shai, Anastassiou et al. 2015), which consists of sodium, potassium, and calcium channels situated on both the soma and dendrites. To activate synaptic currents, we introduced 400 Poisson trains operating at a frequency of 2.5 Hz between 80 ms and 400 ms, which encompassed both AMPA and NMDA currents. We also applied a 0.5 nA current to the soma, which persisted for 100 ms (Supplementary Fig. 7). Please note that we selected a substantial number of random synaptic inputs because it is commonly presumed that the numerical accuracy of a detailed dendritic model is better evaluated by massive random synaptic events.

Results

Our findings suggest that DeepDendrite can effectively reproduce the dendritic and somatic responses of a neuron with excitable dendrites, including dendritic spike generation that may result from synaptic clustering (Supplementary Fig. 7). Notably, the dendritic and somatic responses, including dendritic spikes and somatic bursting, generated by DeepDendrite were identical to those produced by NEURON, indicating that our simulation tool can accurately capture the dynamics of a neuron with excitable dendrites (Supplementary Fig. 7).

Revisions

We have added the contents about numerical accuracy at the end of the section “*Speeding Up DHS by GPU Memory Boosting*”:

“Moreover, compared to the conventional serial Hines method in NEURON running with a single-thread of CPU, DHS speeds up the simulation by 2-3 orders of magnitude (Supplementary Fig. 3), while retaining the identical numerical accuracy in the presence of dense spines (Supplementary Figs. 4 and 8), active dendrites (Supplementary Fig. 7) or different segmentation strategies (Supplementary Fig. 7).”

4. Lines 353-354: *The authors state that they use the same hyperparameters. How do they ensure that the ANN has the same activation function(s) as the biophysical models? For the biophysical models, one can assume that the activation function is the input-output curve (see Fig. 5D). In ANNs this is not clear. Did the authors use a sigmoid function centered at zero?*

Response:

We thank the reviewer to raise important questions regarding the comparison of activation functions between our proposed BDNNs and traditional ANNs. In order to ensure a fairer comparison between the BDNNs and the ANNs, our aim is to make the dynamics of the biophysical model closely resemble that of traditional ANNs. We achieved this by two steps:

First, we set the synaptic currents in BDNNs using the same formula as in traditional ANNs. Specifically, we set the synaptic current triggered by each former-layer neuron as $I_i = g_i * w_i * f(v_i)$, where I_i is the synaptic current triggered by neuron i in the former layer, w_i is the synaptic weight for neuron i in the former layer, f is the activation function at the soma, v_i is the somatic voltage of neuron i in the former layer, g_i is the synaptic conductance. At steady state the transferred synaptic current at soma $I_{soma,i} = r_i * g_i * w_i * f(v_i)$, where r_i is the transfer resistance from the synaptic location to soma. So if viewing $w_i' = r_i * g_i * w_i$, we have $I_{soma,i} = w_i' * f(v_i)$. This setting of synaptic currents has the same formula as the linear part of traditional ANNs: $z_j = \sum_i w_{ij} f(z_i)$, where z_j is the linear response of the node j , and f is the activation function. For computational convenience, we set the resting potential of the biophysical model to 0.

Next, we modified the activation function at soma as follows: at each time step if $v_i \leq 0$, we set $f(v_i) = 0$; if $v_i > 0$, $f(v_i) = v_i$. This is equivalent to the ReLU activation function used in traditional ANNs: $ReLU(z_i) = z_i$ if $z_i > 0$, $ReLU(z_i) = 0$ if $z_i < 0$. Another advantage for such setting of the synaptic currents is the convenience for computing the gradient of each synaptic input.

To make our description of the somatic activation function more rigorous, we've modified

the description as “ReLU-like” in the Method part. We’ve also clarified the definitions in equations (4) and (5).

5. Lines 373-374: *The authors state that dendrites are the key mediator of improved performance under adversarial attacks, which is a conclusion reached after comparing ANNs with their HPC-Net. But a clear explanation of this finding is missing and is rather unintuitive given that dendrites are modelled as passive cables (e.g., see Chavlis and Poirazi, 2021; Acharyaa et al, 2021). Which dendritic feature improves robustness to adversarial attacks? And how would the simulation of active dendritic mechanisms change these findings? I would suggest presenting at least one example/comparison using a biophysical model with active dendritic conductances.*

Response:

We appreciate the critical concerns raised by the reviewer and would like to clarify our focus and provide additional insights into the mechanisms underlying our proposed BDNNs defending against adversarial attacks.

Rationale and hypotheses for using passive dendrites in BDNNs

While our study primarily centers on the development and application of the DHS method and DeepDendrite framework, the neural networks with morphologically detailed passive dendrites serve as a proof-of-concept to showcase the potential of our proposed methodology and framework in advancing state-of-the-art brain-like AI systems. As such, we did not delve deeply into the underlying mechanisms of the observed improvements in the robustness of BDNNs with passive dendrites against adversarial attacks. Nonetheless, we will offer an intuitive explanation based on classical Cable theory and our observations, as well as perform additional experiments to support our claims.

We hypothesize that the spatiotemporal filtering feature of passive dendrites improves the robustness of BDNNs to adversarial attacks. In the experiment settings, we converted input images to spike trains and attached them to the neurons, then added adversarial noise to “perturb” the spike timing. As dendrites are typically modeled as passive cables, Cable theory predicts that their ability to filter dendritic responses during propagation to the soma renders them less susceptible to the “perturbations” in spike timing induced by (adversarial) noise (London and Häusser 2005, Poirazi and Papoutsi 2020). Specifically, our key learning algorithms (Eq. 8 And Eq.9 in the main manuscript, as below) incorporate this concept:

$$W_{ijk} = W_{ijk} - \eta * \frac{dt}{t_e - t_s} \sum_{t=t_s}^{t_e} \Delta W_{ijk}^t$$

$$\Delta W_{ijk}^t = \frac{\partial Loss}{\partial \bar{v}_j} * r_{ijk} * g_{ijk} * f(v_i^t)$$

Here, r_{ijk} is the transfer impedance between the neuron j 's dendrite to the soma computed by NEURON, which may contribute to the filtering of “perturbations”. This feature is a unique aspect in our learning algorithms due to the inclusion of passive dendrites (cables).

Based on this hypothesis, we further predict that placing synaptic inputs located more distally will increase the BDNN's ability to resist adversarial attacks.

Experiments & Results

To test this hypothesis, we compared the robustness of two BDNNs with different connectivity patterns: one with the input neurons connected to the proximal dendrites (< 100 μm) of the hidden neurons, and the other with the input neurons connected to the distal dendrites (> 200 μm) of the hidden neurons (Supplementary Fig. 9a). We also extended the number of training epochs to 60. Apart from these, other model settings as well as training and evaluating procedures are the same as Fig. 6d in the manuscript. The classification accuracy for the two BDNNs as well as the corresponding ANN at different FGSM adversarial noise levels is as in Supplementary Fig. 9b. The BDNNs reached almost the same accuracy as ANN on clean examples (attack strength $\epsilon=0$: 97.23% for proximal, 97.25% for distal, and 97.65% for ANN). As attack strength increased, the accuracy for ANN soon began to drop, while for BDNNs the accuracy almost remained held. Under large attack strength, the accuracy for proximally- and distally-connected networks began to diverge, with 88.65% and 92.96% at attack strength $\epsilon=0.2$ respectively, while still outperforming ANN (78.40%) by a large margin. Our results indicated that the distally-connected model is more robust against large adversarial noise than the proximally-connected model, suggesting that passive dendrites, together with structured synaptic connectivity, can potentially make dendritic neural networks more robust in noisy environments.

Discussions on the path to using active dendrites in BDNNs

Regarding the suggestions of exploring active dendritic mechanisms, we agree that this mission is a critical area of research. However, we could not explore the active dendrite in the current work due to the absence of a highly efficient learning method for active dendrites in biophysically detailed neuron models to perform image classification tasks. Despite recent advancements in dendritic neuron models and their application to AI tasks, most SOTA approaches rely on simplified dendritic models that only contain 2-3 compartments of passive dendrites (Guerguiev, Lillicrap et al. 2017, Sacramento, Costa et

al. 2018, Payeur, Guerguiev et al. 2021). Additionally, the learning methods proposed for morphologically detailed neurons with active mechanisms have been primarily limited to single neurons and have yet to be applied to large-scale networks in relatively challenging tasks, e.g., classical image classification (Bicknell and Häusser 2021, Moldwin, Kalmenson et al. 2021).

It's worth noting that the overall performance of the proposed BDNN models is a combined effect of *architecture design, learning algorithms, training techniques, and active/passive dendritic mechanisms*. There remain numerous challenges, such as how to better combine training techniques (e.g., *hyperparameter tuning, input preprocessing, choice of the optimizer*, etc.) with active or passive dendritic mechanisms. Therefore, extending the learning method from single neurons to large-scale neural networks with active dendrites requires additional significant algorithmic innovations and technical improvements in the training framework, which falls outside the scope of our current work.

Nonetheless, we firmly believe that our utilization of morphologically detailed neural networks with passive dendrites represents a substantial step forward compared to current dendritic ANNs that rely on simplified passive dendrites. The inclusion of additional biophysical properties, such as transfer impedance in the Cable theory and dendritic filtering mechanisms in this work, significantly enhances the computation capacity of single neurons and leads to better performance and robustness of the neural networks as we've shown in the results. Moreover, the morphologically detailed model of passive dendrites is much more biologically realistic than previous simplified dendritic models, allowing for the exploration of other unique biological mechanisms and phenomena, such as brain-wide morphological patterns, and structured synaptic connectivity that we just demonstrated.

The field of biologically-inspired AI with dendritic neurons is rapidly evolving, and we recognize the great potential for future research to investigate the benefits of active dendrites in BDNNs. We plan to explore this area in future work, particularly once suitable learning methods for active dendrites are available. Such advancements could provide further insight into the role of dendrites in improving the robustness of neural networks.

Conclusions

We have clarified that our study primarily focuses on the DHS method and DeepDendrite framework, using neural networks with morphologically detailed passive dendrites as a proof-of-concept. Our experiments suggest that the spatiotemporal filtering feature of passive dendrites, combined with structured synaptic connectivity, can potentially improve the robustness of neural networks against adversarial attacks. However, extending our work to active dendrites requires significant algorithmic innovations and technical

improvements, which we plan to explore in future research.

Revisions

we have added the following statement at the end of the section “*DHS Creates Cell-type-specific Optimal Partitioning*” to clarify our intention:

“It is important to emphasize that the primary goal of employing biophysically detailed dendritic neural networks in AI tasks is not to replicate the success of ANNs. Instead, it aims to provide a proof-of-concept demonstration of the potential advantages of dendritic learning in AI by drawing upon well-established ANN theory.”

we have added the additional experiment and results to address the role of passive dendrites in defending against adversarial attacks in the Discussion part:

“In addition, we investigated the roles passive dendrites play in resisting adversarial attacks. We hypothesized that passive dendrites could serve as spatiotemporal filters, with distal dendrites being more robust to noise interference than proximal dendrites when signals are propagating along the dendrite. To investigate this, we compared two HPC-Nets with distinct connectivity patterns: one with input neurons connected to the proximal part of basal dendrites ($<100\ \mu\text{m}$) and another to the distal part ($>200\ \mu\text{m}$) (Supplementary Fig. 9a). Our results revealed that both HPC-Nets demonstrated greater robustness against adversarial noise than the regular ANN, with the distally-connected model (92.96% accuracy, attack strength $\epsilon=0.2$) outperforming the proximally-connected model (88.65% accuracy, $\epsilon=0.2$) (Supplementary Fig. 9b). This finding suggests passive dendrites may indeed play a role in defending against adversarial attacks by acting as spatiotemporal filters.”

6. Importantly, other studies have also shown the impact of dendrites in improving performance on computer vision tasks. For example, Jones and Kording, 2021; Iyer et al., 2022; Bird et al., 2021, etc. I suggest citing and discussing these studies in the manuscript, especially with respect to interpretability of their improved performance findings.

Response:

We thank the reviewer's suggestion to incorporate and discuss these important studies on the role of active dendrites in improving performance on computer vision tasks. It is crucial to note that most of these models are based on the "point neuron" concept. As such, the parameter settings, network connections, and learning strategies in these studies are “abstractions” derived from spatially extended neurons, such as the sparseness of synaptic inputs or nonlinear response functions of active dendrites. However, detailed biological properties, particularly the spatial information of dendrites, are less exploited in these

studies.

Revisions

We have added the following revisions in the discussion part:

“There has also been progress in the role of active dendrites in ANNs for computer vision tasks. Iyer et al. (2022) proposed a novel ANN architecture with active dendrites, demonstrating competitive results in multi-task and continual learning (Iyer, Grewal et al. 2022). Jones and Kording (2021) used a binary tree to approximate dendrite branching and provided valuable insights into the influence of tree structure on single neurons' computational capacity (Jones and Kording 2021). Bird et al. (2021) proposed a dendritic normalization rule based on biophysical behavior, offering an interesting perspective on the contribution of dendritic arbor structure to computation (Bird, Jedlicka et al. 2021). While these studies offer valuable insights, they primarily rely on abstractions derived from spatially extended neurons, and do not fully exploit the detailed biological properties and spatial information of dendrites. Further investigation is needed to unveil the potential of leveraging more realistic neuron models for understanding the shared mechanisms underlying brain computation and deep learning.”

Minor comments

1. Line 73, Introduction: I suggest adding a verb before “bridge” or using the expression “bridge the gap”, i.e., “[...] were utilized to be the bridge between [...]” or “[...] were utilized to bridge the gap between [...]”.

2. Line 69-70: Please add “[...] such as ion channels, excitatory/inhibitory synaptic currents, etc. [...]” for clarity.

Response:

We have modified the sentences mentioned in the reviewer's comments to improve clarity.

3. Line 79: How do you define “general intelligence”? I suggest adding a sentence with some references in specific domains where AI performs poorly on multiple tasks or exhibits catastrophic forgetting.

Response:

We cited studies reporting catastrophic forgetting in multi-task learning performed by standard ANNs to illustrate the limitation of traditional ANNs compared to the human brain, thus concretizing the definition of “general intelligence”.

4. *Figure S1: I suggest adding the y-axis label “Computational Cost” in both the top and bottom left panels and the x-axis label in the middle of the top row panels.*

Response:

We modified figure S1 as the reviewer suggested.

5. *Lines 167, 227, 252, etc.: Please add a comma after “i.e.” or “e.g.” such that “i.e.,” or “e.g.,” OR remove the comma from such statements for consistency across the manuscript.*

6. *Line 174: Please write the “k” in “k parallel threads” with italic as in line 184 for consistency.*

Response:

We corrected these inconsistent notations in our manuscript.

7. *Line 193: Please cite the ModelDB as: “McDougal RA, Morse TM, Carnevale T, Marengo L, Wang R, Migliore M, Miller PL, Shepherd GM, Hines ML. Twenty years of ModelDB and beyond: building essential modeling tools for the future of neuroscience. Journal of Computational Neuroscience 2017; 42(1):1-10 (2017).”. See here for more details: [https://senselab.med.yale.edu/modeldb/HowToCite#:~:text=When you use ModelDB to,The ModelDB \(McDougal et al.](https://senselab.med.yale.edu/modeldb/HowToCite#:~:text=When you use ModelDB to,The ModelDB (McDougal et al.)*

Response:

We corrected the citation format of ModelDB according to <https://senselab.med.yale.edu/modeldb/HowToCite>.

8. *Line 307: The field of “deep learning” consists of more architectures than DNNs. I suggest rephrasing the sentence.*

Response:

We rephrased this sentence and clarified the relationship between deep learning and DNNs.

9. *Lines 312-321: Please add respective references for the “recent work” (line 312) and the “foundational work” (line 315) mentioned in the text.*

10. *Line 451: Please add “[...] increase the internal [Ca²⁺]”. The superscript “2+” should be placed inside the squared brackets.*

11. *Line 509: Please add some indicative references for the “recent studies” mentioned here.*

Response:

We added references omitted in the manuscript and corrected notation mistakes according to the reviewer's comments.

Reviewer #3 (Remarks to the Author):

"I find the results of the adversarial attacks promising (Figure 6e), but feel that more analysis would need to be done in order to be convinced that dendritic computations confer such robustness (otherwise, I think such claims should be toned down in the manuscript)."

We thank the reviewer for considering our results "promising" and appreciate the suggestions regarding validations of the results of adversarial attacks. We would like to clarify that the results of the adversarial attacks in the proposed neural networks with morphologically detailed neurons act as proof-of-concept to illustrate the viability and potential of our proposed framework for building more brain-like AI models with biophysically detailed neural networks (BDNNs). Therefore, we do not intend to claim that our results are conclusive or comprehensive in this regard. Before addressing the reviewer's concerns, we would like to provide some background information on performing AI tasks with dendritic models, as well as highlight our main technical contributions in developing DeepDendrite.

Background

In contrast to the simplified dendritic models, which can be directly implemented in popular deep learning frameworks such as PyTorch (Paszke, Gross et al. 2019) and TensorFlow (Abadi, Barham et al. 2016), training biophysically detailed networks for contemporary AI tasks poses tremendous technical challenges. A novel learning framework must be developed on top of domain-specific simulation platforms, such as NEURON, to ensure numerical accuracy in simulations and enable the training of biophysically detailed networks. Moreover, the computational cost of training biophysically detailed networks is orders of magnitude greater than that of simplified dendritic models, making it difficult to train networks of a certain scale.

Technical contributions

In this context, DeepDendrite is an important technological leap forward that addresses these challenges:

1. We handcrafted the entire basic deep learning module from scratch in DeepDendrite, including *high-level layers API, automatic gradient computation, GPU support, mini-batch training, input pipeline, SGD/Adam optimizer, and more.*
2. Our entire framework is written in CUDA, ensuring high computational efficiency on GPUs and allowing for easy parallelization and scaling to larger datasets and models.
3. We made our own deep learning module fully compatible with the modified NEURON (CoreNEURON is its GPU-version) simulator, allowing them to exchange information during run-time. Note that simulations in the original NEURON simulator cannot be disrupted and thus our modified simulator makes it possible to interrupt the simulations at any point for training purposes.
4. We integrated our DHS method into the NEURON solver, reducing the training time on MNIST from weeks to hours and enabling the training of biophysically detailed networks on modern AI tasks.
5. We proposed a novel learning method for fast training of detailed networks, which has not been previously explored in the literature. To the best of our knowledge, our study is the first to demonstrate the feasibility of training biophysically detailed networks on modern AI tasks, with promising results in comparison to ANNs.

Regarding the reviewer's suggestion to validate the results of adversarial attacks, we will include additional experiments on adversarial attacks to further validate the effectiveness of our proposed methodology and framework, and will revise our conclusions accordingly.

This ultimately boils down to recommending the following:

1. *More challenging, ecologically plausible visual task (e.g. ImageNet)*
 2. *Dendrites implanted into deeper networks (that are known to match primate visual cortex, e.g. ResNet-18)*
 3. *More networks through which the adversarial examples are generated from*
 4. *More methods of generating adversarial attacks beyond FGSM (which are easily testable since you used the Bethge Lab Foolbox library)*
- 1-3 are important to demonstrate at a minimum, and having 4 on ImageNet would *really* strengthen the result."*

Specifically, the reasoning for points 1 & 2 come from the following observations:

In lines 351-352, the base network of comparison is too simple, given that the MNIST performance is low (good networks get at least 99% on this task). With that desired goal of 99% accuracy in mind, it is also clear that this is too simple of a task to separate models, especially when making claims of biological plausibility. This is evident, for example, with biologically plausible learning rules, where training on MNIST with feedback alignment leads to similar performance as

backpropagation, but is clearly a suboptimal choice on ImageNet (Bartunov et al. NeurIPS 2018). Furthermore, lots of work has shown that training ANNs on ImageNet has yielded neurally plausible representations (Yamins et al. 2014, Hong et al 2016; Khaligh-Razavi and Kriegeskorte 2014, Guclu and van Gerven 2015, and others), underscoring the importance of training on this imageset when making claims of biological plausibility conferring a behavioral benefit (in this case, increased adversarial robustness).

*Not only that, but training on ImageNet should be feasible since you used A100s, and would further underscore the first part of the paper which deals with the efficiency of DeepDendrite. My specific suggestion would be to have ResNet-18 as your base ANN model of comparison, and implant your DeepDendrite into it. I would be curious especially if it could match the performance of deeper models, such as ResNet-50, since dendritic computations tend to have lots of additional nonlinear operations. If that was true, that would *further* strengthen the result (as an additional behavioral metric beyond adversarial robustness), and if not that is fine too, so long as adversarial robustness is maintained still on this imageset.*

We thank the reviewer for their insightful suggestion to validate our results on the ImageNet dataset. We agree that MNIST is a relatively simple dataset for evaluating the performance and robustness of different models. However, we would like to reiterate that our primary goal of applying BDNNs to classification tasks on MNIST or other classic datasets is not to replicate ANNs' success or to claim superior performance. Instead, we aim to use ANNs as a starting point and explore the uncharted potentials of dendritic learning in AI tasks by anchoring on the well-established theory and experience of ANNs. By doing so, we seek to make BDNNs more closely resemble our cognitive systems than to mimic ANNs.

Discussions on the uniqueness of BDNNs

We would also like to emphasize that BDNNs are fundamentally different from conventional ANNs or ANNs with biologically plausible algorithms in terms of their unique neurodynamic system. Unlike ANNs or ANNs trained by biologically plausible algorithms, where neurons are modeled as point-like entities with static activation functions such as ReLU, BDNNs are generally modeled with hundreds to thousands of ordinary differential equations (ODEs) that describe the dynamics of dendritic and somatic integration. This is a significant departure from the traditional neuron model in ANNs, which opens up new research questions and opportunities to study the interplay between neural dynamics and AI.

Despite our extensive knowledge of dendrites in the brain, their role in AI tasks is still largely unexplored. Only recently has some seminal work shed light on this topic, such as the segregated feedforward and feedback pathways in simplified dendritic neuron models that enable simultaneous feedforward computation and error-backpropagation (Guerguiev, Lillicrap et al. 2017, Naud and Sprekeler 2018, Sacramento, Costa et al. 2018, Richards

and Lillicrap 2019, Payeur, Guerguiev et al. 2021). While not explicitly investigated in this work, certain aspects of the segregated pathways are incorporated into our proposed learning algorithms. We posit that these learning algorithms can be further refined in future research to more closely resemble the parallel top-down and bottom-up processes present in our cognitive system. This serves as an example of how BDNNs hold the potential to surpass conventional ANNs in certain aspects.

Response to the reviewer's concerns

We have made significant improvements to the DeepDendrite framework and conducted additional experiments to enhance the robustness of our results. We have organized our response into six sections, addressing each critical point raised by the reviewer:

1. DeepDendrite Framework Improvements:

In the past six months, our team has dedicated significant effort to enhance the DeepDendrite framework. Our efforts have resulted in several key technical improvements that enable the training of larger and deeper detailed dendritic networks, including:

- 1) *A separate forward-backward pathway method* that assigns a single-compartment “mirror neuron” to each forward neuron to compute its somatic gradient. This boosts the gradient flow and reduces signal interference.
- 2) *A modular code architecture* that eases building deep detailed networks with our layers API and establishing the backward pathway with minimal code changes.
- 3) *A refined synaptic weight learning rule* that avoids instability due to averaging the gradient at each time step because of voltage fluctuations.

2. Challenges in ImageNet and Implementing ResNet-18/ResNet-50

Despite these improvements, we have encountered challenges in scaling our models for multi-GPU training due to our limited financial and human resources. As previously mentioned, we have to handcraft the basic deep learning module ourselves, without leveraging popular platforms such as PyTorch or TensorFlow, due to compatibility constraints of the NEURON simulator. These platforms have undergone extensive optimization by industry leaders, with tremendous investments in both financial and human resources, incorporating many tricks essential for improving deep learning model efficiency. Therefore, while we have identified the bottleneck as being in the message communication between the learning module and the NEURON simulator in the newly built multi-GPU DeepDendrite, we can only train models on a single GPU at this stage, hindering our ability to tackle the ImageNet dataset.

We attempted to train relatively larger BDNNs with ImageNet dataset on a single GPU.

However, this has proven to be impractical due to the long training time and high memory consumption. To illustrate, DeepDendrite framework has reduced the training time of a three-layer BDNN on the MNIST dataset from 220 hours to about 8 hours. Nonetheless, training with ImageNet, which contains 1.2 million images, is prohibitively expensive due to its increased size and complexity, requiring larger networks and reduced batch size. Likewise, implementing ResNet-18 or ResNet-50 in the DeepDendrite framework is also impractical, as convolution layers are very costly to implement with biophysically detailed neurons.

Given our current resources, training with ImageNet would take at least *three months* on a single A100 GPU, making it impractical to train the network and perform hyperparameter searching. However, we are excited about the future potential of our technology and are confident that we can overcome this limitation by employing dozens or hundreds of A100 GPUs in an improved multi-GPU DeepDendrite framework. With our continued dedication and innovation, we look forward to advancing our promising technology and unlocking its full potential.

3. Experiments on CIFAR-10 and MNIST Datasets with deeper BDNNs

Despite these limitations, we have conducted additional experiments with deeper BDNNs on CIFAR-10 and MNIST datasets to demonstrate the capabilities of our detailed network models and learning methods. These experiments enabled us to evaluate the models' performance within a reasonable timeframe while still leveraging the recent technical improvements we made to the DeepDendrite framework.

Experimental Settings: Specifically, our original learning rule (see Methods equations 8 and 9) averages the gradient at each time step during learning, we found such a setting occasionally leads to unstable results since the voltage at each time step can vary rapidly. So we've come up with the modified weight update rule as shown in Supplementary Equation 1:

$$W_{ijk} = W_{ijk} - \eta * \frac{\partial Loss}{\partial \bar{v}_j} * r_{ijk} * g_{ijk} * f(\bar{v}_i) \quad (S1)$$

Here W_{ijk} is the synaptic weight of the k^{th} synapse connecting from neuron i to neuron j . η is the learning rate. \bar{v}_i, \bar{v}_j are the average somatic voltages of neuron i and j during the learning period respectively, i.e., $\bar{v}_i = \frac{dt}{t_e - t_s} \sum_{t=t_s}^{t_e} v_i^t$, where t_s, t_e are the start time and end time for learning respectively, v_i^t is the somatic voltage of neuron i at time step t . r_{ijk} is the transfer resistance from the synaptic location on neuron j 's dendrite to its soma. g_{ijk} is the synaptic conductance, and f is the activation function at soma. The error term $\frac{\partial Loss}{\partial \bar{v}}$ is

calculated by applying the chain rule,

$$\frac{\partial Loss}{\partial \bar{v}_i^l} = \sum_{j=0}^{N_{l+1}} \frac{\partial Loss}{\partial \bar{v}_j^{l+1}} * r_{ijk} * g_{ijk} * W_{ijk} * f'(\bar{v}_i^l) \quad (S2)$$

Here N_{l+1} is the number of neurons that neuron i at layer l connects to at layer $l+1$. During implementation in order to coop with the simulator's computational logic, $\frac{\partial Loss}{\partial \bar{v}}$ is approximately calculated by the voltage of single-compartment mirror neurons. Specifically, we set the synaptic current from neuron j 's mirror neuron to neuron i 's mirror neuron to be

$$I_{ji} = g_m * \frac{\partial Loss}{\partial \bar{v}_j^{l+1}} * r_{ijk} * g_{ijk} * W_{ijk} * f'(\bar{v}_i^l) \quad (S3)$$

where g_m is the mirror neuron's synaptic conductance. At steady state the voltage of neuron i 's mirror neuron equals to $\sum_{j=0}^{N_{l+1}} r_m * I_{ji}$, where r_m is the mirror neuron's input resistance.

So by setting $g_m = 1/r_m$, Supplementary Equation 2 can be approximated by the steady state voltage of neuron i 's mirror neuron.

After refining our architectures, algorithms, and tools, we conducted experiments on larger networks. We discovered that the original simulation time of 50 ms per image was insufficient to drive responses in deeper layers, prompting us to extend the simulation time to 500 ms per image, with learning beginning at $t_s = 200$ ms and ending at $t_e = 500$ ms. To keep the total training time affordable, we also extended the simulation time step.

Next, following the review's recommendation of Bartunov et al. *NeurIPS* (2018), we constructed five networks with the same architectures (two fully-connected networks for MNIST and CIFAR-10, and three convolutional networks for MNIST, CIFAR-10, and ImageNet). Notably, the current fully-connected BDNN for CIFAR-10 is approximately 50 times larger than the previous 3-layer HPC-Net.

Results: We estimated the training time required for these networks and discovered that, even when utilizing DeepDendrite on a single A100 GPU, the training duration for convolutional BDNNs was prohibitively long. Specifically, for MNIST and CIFAR-10, completing 60 epochs of training was estimated to take two and four weeks, respectively. Due to the excessive training time, we could not complete even a single epoch of training for the ImageNet network. As a result, we shifted our focus to the two fully-connected BDNNs for MNIST and CIFAR-10 (Supplementary Figs. 10a and 11a, for MNIST and CIFAR-10 BDNNs, respectively), which required three days and two weeks, respectively, to complete 60 epochs of training.

We observed an intriguing phenomenon where the model's robustness appears to be

closely related to “structured connectivity”, a key concept in dendritic computation, referring to the organization of synaptic inputs on dendrites in a specific manner that enables complex computations (Druckmann, Feng et al. 2014, Poirazi and Papoutsis 2020). We found that the strategic placement of synapses in distal dendrites significantly enhanced robustness in the smaller-sized BDNN (Supplementary Fig. 9b). From an intuitive standpoint, intricate dendrites, modeled as passive cables, serve as spatiotemporal filters, smoothing fluctuating signals and thereby alleviating signal “perturbations” when signals are propagating along the dendrite (London and Häusser 2005, Poirazi and Papoutsis 2020). The longer the dendrite (or “cable”), the better it will be at filtering noise signals. Accordingly, instead of randomly distributing synapses in the dendrite, we positioned the feedforward synapses onto the distal dendrites of the two deeper fully-connected BDNNs, which hopefully contributed to the adversarial robustness of the larger and deeper BDNNs. Further research is necessary to explore in-depth the optimal strategy for synaptic placement in dendrites to enhance BDNNs performance and robustness.

The experiment details are summarized in Supplementary Table 2. The learning rates were carefully selected for the ANNs to produce the highest clean test accuracy at the end of training, then we applied the same learning rates on BDNNs respectively. Unless otherwise noted, other parameters and training settings for the BDNNs were identical to those employed for the original 3-layer HPC-Net.

Our experiments demonstrated a significant increase in training speed for the MNIST BDNN and CIFAR-10 BDNN, achieving a 45-fold and 57-fold speedup, respectively, when compared to the same training procedure with 40-process-parallel NEURON on a single CPU (Intel Xeon E5-2698 v4), utilizing a single GPU (NVIDIA Tesla A100) (Supplementary Figs. 10b and 11b, for MNIST and CIFAR-10, respectively). These results highlight the potential of DeepDendrite's improved framework for enhancing the training efficiency of BDNNs.

Our results also indicated that, for the fully-connected architectures examined in this study, BDNNs achieved nearly identical or comparable clean classification accuracy to their corresponding ANNs (Supplementary Fig. 10c, for MNIST, attack strength at $\epsilon=0$: BDNN 98.29%, ANN 98.29%; and Supplementary Fig. 11c for CIFAR-10, $\epsilon=0$: BDNN 52.70%, ANN 53.13%). It is important to note that the BDNNs were trained using our custom-built framework, while all ANNs were trained on the highly-optimized platform PyTorch. We are very proud that the custom-built framework for training BDNNs is effective and can be further optimized to potentially achieve even better results.

Conclusions: We have made significant improvements to the DeepDendrite framework,

enabling the training of larger and deeper detailed dendritic networks. However, due to resource limitations and compatibility constraints with the NEURON simulator, we have faced challenges in scaling our models for multi-GPU training. Despite these limitations, our experiments on CIFAR-10 and MNIST datasets demonstrate that BDNNs can achieve nearly identical or comparable clean test accuracy to their corresponding ANNs. In addition, we observed a correlation between the model's robustness and “structured connectivity”, potentially offering a promising direction for enhancing BDNNs performance and robustness.

4. Expanded Experiments with Multiple Base Models and Attack Methods

Experiments: By incorporating the reviewer's valuable suggestions, we have greatly enriched our study by successfully training a VGG (Simonyan and Zisserman 2015) with 11 weight layers as an additional base model and including additional attack methods. The VGG model achieved a test accuracy of 99.53% on MNIST and 84.18% on CIFAR-10. To diversify attack methods, we incorporated 5-step PGD (Madry, Makelov et al. 2018) and DeepFool (Moosavi-Dezfooli, Fawzi et al. 2016) in addition to the existing FGSM. Consequently, we employed 2 base models (the 20-layer ResNet from our manuscript and the newly trained VGG) and 3 attack methods to generate adversarial samples, resulting in 6 combinations for both MNIST and CIFAR-10.

Results: We subsequently assessed the prediction accuracy of BDNNs and ANNs, as previously described, by feeding them the adversarial samples (Fig. 6d). Our findings highlight the remarkable capability of BDNNs to defend against transfer attacks, outperforming ANNs across all combinations (Supplementary Figs. 10c and 11c, for MNIST and CIFAR-10, respectively). For more successful attacks, BDNNs achieve an accuracy improvement of up to 5.6% for MNIST and 5.5% for CIFAR-10 under high attack strength (Supplementary Fig. 10c, for MNIST, FGSM on ResNet20 at $\epsilon=0.2$; and Supplementary Fig. 11c, for CIFAR-10, PGD on ResNet20 at $\epsilon=0.36$). Although we have not yet experimented with convolutional BDNNs or ImageNet, we eagerly anticipate pursuing these directions in the near future to enhance the robustness of our results and further explore the potential of BDNNs.

Conclusions: In response to reviewer suggestions, we expanded our experiments by using multiple base models (ResNet, VGG) and incorporating multiple attack methods (FGSM, PGD, and DeepFool) to generate transferred examples, resulting in six combinations for MNIST and CIFAR-10. Our findings showcase the impressive performance of BDNNs in consistently outperforming ANNs when defending against transfer attacks across all tested combinations.

5. Future directions and Limitations

While we have achieved promising results, we recognize the exciting potential for further exploration and addressing limitations in the following areas:

- a) *Scalability and Multi-GPU Systems*: The current experiments were limited in scope and computational power. As we eagerly take on more challenging datasets like ImageNet, we are committed to developing scalable BDNNs that can effectively leverage multi-GPU systems. Future efforts should be dedicated to solving the communication problems that have been identified in the current multi-GPU framework.
- b) *Resembling Human Cognitive System*: please keep in mind that the primary goal of BDNNs is NOT to replicate ANN's success, but to resemble our cognitive systems. Therefore, future research should focus on incorporating biological and cognitive principles into the design of BDNNs. These principles may include parallel top-down and bottom-up pathways, active dendrites, structured connectivity, etc., which improve BDNNs' performance and resilience in "noisy" environment.
- c) *Structured connectivity*: one intriguing finding in our experiment is that the BDNN's performance and robustness is sensitive to the exact location of synapses in the dendrite. Although due to the long training time, we could not explore the optimal strategy for synaptic placement in deeper and larger BDNNs. Future research can perform extensive hyper-parameter searching for the optimal locations of feedforward synapses by leveraging the improved multi-GPU DeepDendrite framework's power.
- d) *Applications in Natural Language Processing (NLP)*: While our current research has successfully tackled image classification tasks, we are excited about the potential for applying BDNNs to NLP tasks, broadening their applicability. The inherently noisy and ambiguous nature of human language may make BDNNs particularly well-suited for NLP tasks, as they could potentially offer increased robustness against adversarial examples and improved performance in handling ambiguous or uncertain linguistic inputs. Future research will explore the potential of BDNNs in NLP tasks, as well as the challenges and limitations associated with scaling them to larger datasets.
- e) *Limitations*: The scalability of BDNNs in larger-scale problems remains to be tested, and their performance across a wide range of tasks and domains is not yet fully understood. Furthermore, the incorporation of biological principles

into the design of BDNNs such as active dendrites may introduce additional computational complexity, which could impact training and inference time. These limitations must be carefully considered and addressed in future research.

6. Revisions to the manuscripts

Some of the experiment and results details are included in Supplementary Materials.

We have added the following statement at the end of the section, “*DHS Creates Cell-type-specific Optimal Partitioning*”, to clarify our intention:

“It is important to emphasize that the primary goal of employing biophysically detailed dendritic neural networks in AI tasks is not to replicate the success of ANNs. Instead, it aims to provide a proof-of-concept demonstration of the potential advantages of dendritic learning in AI by drawing upon well-established ANN theory.”

We have taken the reviewer's feedback into account and added the following content in the Results section, “*DHS-based Learning on Detailed Dendritic Neural Networks*”, to highlight the progress and achievements of our work:

“Finally, we extended our experiments to deeper and larger HPC-Nets by moving to a more challenging dataset (CIFAR-10), training an additional base model (VGG), and incorporating additional attack methods (PGD and DeepFool), resulting in 6 combinations for both MNIST (Supplementary Fig. 10) and CIFAR-10 (Supplementary Fig. 11). Our findings show that HPC-Nets consistently outperform ANNs in defending against transfer attacks, with accuracy improvements of up to 5.6% for MNIST and 5.5% for CIFAR-10 under high attack strength (Supplementary Figs. 10 and 11).”

We also agree with the reviewer that our conclusions should be more rigorous, therefore, we add the following statement in the same paragraph:

“Our results are the first to show that dendrites might be crucial for biological vision robustness. Further experiments are required to examine these findings in more challenging datasets such as ImageNet (Bartunov et al. 2018).”

We add the following content in the last paragraph of the Discussion, to address the future directions and limitations of our current work:

“In conclusion, DeepDendrite has shown remarkable potential in image classification tasks, opening up a world of exciting future directions and possibilities. To further advance DeepDendrite and the application of biologically detailed dendritic models in AI tasks, we may focus on emulating human cognitive systems, developing multi-GPU systems, and exploring applications in other domains, such as Natural Language

Processing (NLP), where dendritic filtering properties align well with the inherently noisy and ambiguous nature of human language. Challenges include testing scalability in larger-scale problems, understanding performance across various tasks and domains, and addressing the computational complexity introduced by novel biological principles, such as active dendrites. By overcoming these limitations, we can further advance the understanding and capabilities of biophysically detailed dendritic neural networks, potentially uncovering new advantages, enhancing their robustness against adversarial attacks and noisy inputs, and ultimately bridging the gap between neuroscience and modern AI.”

Parameter	BDNNs (MNIST)	BDNNs (CIFAR-10)	ANNs
Platform	DeepDendrite	DeepDendrite	PyTorch
Simulation time step (ms)	5	5	N/A
Input spike interval (ms)	$50 / (p(x, y, c) + 0.01)$	$50 / (p(x, y, c) + 0.01)$	N/A
Input spike start (ms)	$90 + 50 / (p(x, y, c) + 0.01)$	$90 + 50 / (p(x, y, c) + 0.01)$	N/A
Input synaptic reversal potential (mV)	1	1	N/A
Input synaptic time constant (ms)	5	5	N/A
Input neuron membrane resistance ($\Omega \text{ cm}^2$)	10^5	10^5	N/A
Mirror neuron membrane resistance ($\Omega \text{ cm}^2$)	1	1	N/A
Hidden layer architecture	5 layers each with 256 biophysically detailed neurons	3 layers each with 1024 biophysically detailed neurons	Same as corresponding BDNNs
Activation function	ReLU-like	ReLU-like	ReLU
Synaptic connections per neuron	1	1	1
Weight initializer	Glorot uniform	Glorot uniform	Glorot uniform
Batch size	4	2	Same as corresponding BDNNs
Optimizer	SGD without momentum or weight decay	SGD without momentum or weight decay	SGD without momentum or weight decay
Learning rate	0.005	0.0001	Same as corresponding BDNNs
Total training time (epochs)	60	60	60

Supplementary Table 2 | Summary of experiment details for fully-connected networks. Note: "p(x, y, c)" represents the normalized grayscale or color channel value for MNIST or CIFAR-10, respectively. N/A, not available.

Supplementary Figure 7 | Numerical accuracy of DeepDendrite compared with NEURON under different segmentations. The pyramidal neuron model with active dendrites was taken from (Shai, Anastassiou et al. 2015). The dendritic and somatic responses, including dendritic spikes and somatic bursting, generated by DeepDendrite were identical to those produced by NEURON. **a** Comparison of five voltage traces recorded at soma and four different dendritic sites simulated using DeepDendrite and NEURON under the original segmentation. **b** Comparison of five voltage traces recorded at soma and four different dendritic sites simulated using DeepDendrite and NEURON under the new, more fine-grained segmentation.

Supplementary Figure 8 | Numerical accuracy of DeepDendrite compared with NEURON under different spine densities. The dendritic and somatic responses generated by DeepDendrite were identical to those produced by NEURON. **a** Comparison of five voltage traces recorded at soma and four different dendritic sites simulated using DeepDendrite and NEURON on the full spine model. **b** Comparison of five voltage traces recorded at soma and four different dendritic sites simulated using DeepDendrite and NEURON on the few spine model.

Supplementary Figure 9 | Robustness of HPC-nets with different connectivity patterns. **a** Two connectivity patterns were experimented: one with the input neurons connected to the proximal dendrites ($< 100 \mu\text{m}$) of the hidden neurons, and the other with the input neurons connected to the distal dendrites ($> 200 \mu\text{m}$) of the hidden neurons. **b** Classification accuracy of the two BDNNs as well as the corresponding ANN under transfer attack with different strength levels.

Supplementary Figure 10 | Fully-connected HPC-net trained on MNIST. a Network architecture of the HPC-net with five hidden layers. Only feedforward pathway is shown for clarity. **b** Run time of one epoch training. Parallel NEURON + Python: training on a single CPU with multiple cores, using 40-process-parallel NEURON to simulate the BDNN and extra Python code to support mini-batch training. DeepDendrite: training on a single GPU with DeepDendrite. **c** Classification accuracy of the BDNN and the corresponding ANN on transferred examples generated by different combinations of base models and attack methods.

Supplementary Figure 11 | Fully-connected HPC-net trained on CIFAR-10. a Network architecture of the HPC-net with three hidden layers. Only feedforward pathway is shown for clarity. **b** Run time of one epoch training. Parallel NEURON + Python: training on a single CPU with multiple cores, using 40-process-parallel NEURON to simulate the BDNN and extra Python code to support mini-batch training. DeepDendrite: training on a single GPU with DeepDendrite. **c** Classification accuracy of the BDNN and the corresponding ANN on transferred examples generated by different combinations of base models and attack methods.

References

- Abadi, M., P. Barham, J. M. Chen, Z. F. Chen, A. Davis, J. Dean, M. Devin, S. Ghemawat, G. Irving, M. Isard, M. Kudlur, J. Levenberg, R. Monga, S. Moore, D. G. Murray, B. Steiner, P. Tucker, V. Vasudevan, P. Warden, M. Wicke, Y. Yu and X. Q. Zheng (2016). "TensorFlow: A system for large-scale machine learning." Proceedings of Osd1'16: 12th Usenix Symposium on Operating Systems Design and Implementation: 265-283.
- Bicknell, B. A. and M. Häusser (2021). "A synaptic learning rule for exploiting nonlinear dendritic computation." Neuron **109**(24): 4001-4017. e4010.
- Bird, A. D., P. Jedlicka and H. Cuntz (2021). "Dendritic normalisation improves learning in sparsely connected artificial neural networks." PLOS Computational Biology **17**(8): e1009202.
- Druckmann, S., L. Q. Feng, B. Lee, C. Yook, T. Zhao, J. C. Magee and J. Kim (2014). "Structured Synaptic Connectivity between Hippocampal Regions." Neuron **81**(3): 629-640.
- Guerguiev, J., T. P. Lillicrap and B. A. Richards (2017). "Towards deep learning with segregated dendrites." eLife **6**: e22901.
- Iyer, A., K. Grewal, A. Velu, L. O. Souza, J. Forest and S. Ahmad (2022). "Avoiding catastrophe: Active dendrites enable multi-task learning in dynamic environments." Frontiers in neurorobotics **16**.
- Jones, I. S. and K. P. Kording (2021). "Might a single neuron solve interesting machine learning problems through successive computations on its dendritic tree?" Neural Computation **33**(6): 1554-1571.
- London, M. and M. Häusser (2005). "Dendritic computation." Annual Review of Neuroscience **28**: 503-532.
- Madry, A., A. Makelov, L. Schmidt, D. Tsipras and A. Vladu (2018). Towards Deep Learning Models Resistant to Adversarial Attacks. 6th International Conference on Learning Representations. Y. Bengio and Y. LeCun. Vancouver, BC, Canada: 1-28.
- Moldwin, T., M. Kalmenson and I. Segev (2021). "The gradient clusteron: A model neuron that learns to solve classification tasks via dendritic nonlinearities, structural plasticity, and gradient descent." PLoS computational biology **17**(5): e1009015-e1009015.
- Moosavi-Dezfooli, S. M., A. Fawzi and P. Frossard (2016). "DeepFool: a simple and accurate method to fool deep neural networks." 2016 IEEE Conference on Computer Vision and Pattern Recognition (CVPR): 2574-2582.
- Naud, R. and H. Sprekeler (2018). "Sparse bursts optimize information transmission in a multiplexed neural code." Proceedings of the National Academy of Sciences of the United States of America **115**(27): E6329-E6338.
- Paszke, A., S. Gross, F. Massa, A. Lerer, J. Bradbury, G. Chanan, T. Killeen, Z. M. Lin, N. Gimelshein, L. Antiga, A. Desmaison, A. Kopf, E. Yang, Z. DeVito, M. Raison, A. Tejani, S. Chilamkurthy, B. Steiner, L. Fang, J. J. Bai and S. Chintala (2019). PyTorch: An Imperative Style, High-Performance Deep Learning Library. Advances in Neural Information Processing Systems 32 (NeurIPS 2019). **32**.
- Payeur, A., J. Guerguiev, F. Zenke, B. A. Richards and R. Naud (2021). "Burst-dependent synaptic plasticity can coordinate learning in hierarchical circuits." Nature Neuroscience **24**(7): 1010-1019.
- Poirazi, P. and A. Papoutsi (2020). "Illuminating dendritic function with computational models." Nature Reviews Neuroscience **21**(6): 303-321.
- Richards, B. A. and T. P. Lillicrap (2019). "Dendritic solutions to the credit assignment problem." Current Opinion in Neurobiology **54**: 28-36.

Sacramento, J., R. P. Costa, Y. Bengio and W. Senn (2018). Dendritic cortical microcircuits approximate the backpropagation algorithm. Advances in Neural Information Processing Systems 31 (NeurIPS 2018). **31**.

Shai, A. S., C. A. Anastassiou, M. E. Larkum and C. Koch (2015). "Physiology of layer 5 pyramidal neurons in mouse primary visual cortex: coincidence detection through bursting." PLoS computational biology **11**(3): e1004090.

Simonyan, K. and A. Zisserman (2015). Very Deep Convolutional Networks for Large-Scale Image Recognition. 3rd International Conference on Learning Representations. Y. Bengio and Y. LeCun. San Diego, CA, USA: 1-14.

REVIEWER COMMENTS

Reviewer #1 (Remarks to the Author):

The authors responded to all my comments. I have no further comments.

Reviewer #2 (Remarks to the Author):

The authors have addressed all of my comments and the manuscript has improved greatly. I suggest it is accepted for publication.

Congratulations for the great work!

Reviewer #3 (Remarks to the Author):

I appreciate the good faith effort the authors have put in to run additional experiments to determine the rigor of their result. The authors point out in their rebuttal that the primary goal of BDNNs is not to replicate ANN success, but rather, to resemble our cognitive systems. I completely agree, as this was my motivation for wanting to see performance and adversarial robustness on vision tasks, that we know humans can perform. More generally, a convincing measure of resembling our cognitive systems is to, at a minimum, resemble our behavioral capabilities. After all, neural circuits combine to give rise to complex behaviors that ensure an animal's survival!

While I would have wanted to see results on ImageNet, the author's current results convince me of my earlier concern that training on a harder dataset than MNIST (in this case, CIFAR-10), would diminish the efficacy of their approach. Specifically, in Supplementary Figure 11, across all methods of adversarial attacks, the difference between the red and blue curves is quite small, and in fact, much smaller than on MNIST, whose results are displayed in Supplementary Figure 10 (compare, for example, at epsilon 0.2 the difference in performance between the red and blue curves with a ResNet20 on DeepFool with CIFAR-10 vs. MNIST).

Furthermore, the overall performance of the networks tested on CIFAR-10 is quite low, achieving at best around 52% accuracy, despite hyperparameter tuning of their BDNN, whereas on MNIST, the accuracy is

98%. To me, these additional experiments demonstrate that, as it currently stands, this method is not a convincingly promising framework for "bridging the gap between modern neuroscience and AI". Not to mention, that it appears difficult at the moment to not only implant this into convolutional neural networks (which have been shown in many prior works to yield neurally plausible representations, compared to the fully feedforward networks they try), but also difficulty to scale things up successfully to CIFAR-10, let alone ImageNet.

With these considerations in mind, I cannot recommend this paper for publication.

Dear Reviewers,

Thank you for the valuable time and efforts in reviewing our manuscript and providing insightful comments. Following the recommendations from Reviewer #1, Reviewer #3, and the editor, we have undertaken careful adjustments to our manuscript. Specifically, we have condensed the section concerning the Artificial Neural Network (ANN) into a single figure for a concise discussion, thereby ensuring a clearer focus on our study's primary objective. The Abstract, Introduction, Results, and Discussion sections have also been updated to reflect changes related to the ANN part.

Reviewer #1 (Remarks to the Author):

The authors responded to all my comments. I have no further comments.

Response:

We express our gratitude for your positive evaluation of our work.

Reviewer #2 (Remarks to the Author):

The authors have addressed all of my comments and the manuscript has improved greatly. I suggest it is accepted for publication.

Congratulations for the great work!

Yiota Poirazi

Response:

Thank you for your generous feedback and for recommending our manuscript for publication.

Reviewer #3 (Remarks to the Author):

I appreciate the good faith effort the authors have put in to run additional experiments to determine the rigor of their result. The authors point out in their rebuttal that the primary goal of BDNNs is not to replicate ANN success, but rather, to resemble our cognitive systems. I completely agree, as this was my motivation for wanting to see performance and adversarial robustness on vision tasks, that we know humans can perform. More generally, a convincing measure of resembling our cognitive systems is to, at a minimum, resemble our behavioral capabilities. After all, neural circuits combine to give rise to complex behaviors that ensure an animal's survival!

While I would have wanted to see results on ImageNet, the author's current results convince me of my earlier concern that training on a harder dataset than MNIST (in this case, CIFAR-10), would diminish the efficacy of their approach. Specifically, in Supplementary Figure 11, across all methods of adversarial attacks, the difference between the red and blue curves is quite small, and in fact, much smaller than on MNIST, whose results are displayed in Supplementary Figure 10 (compare, for example, at epsilon 0.2 the difference in performance between the red and blue curves with a ResNet20 on DeepFool with CIFAR-10 vs. MNIST).

Furthermore, the overall performance of the networks tested on CIFAR-10 is quite low, achieving at best around 52% accuracy, despite hyperparameter tuning of their BDNN, whereas on MNIST, the accuracy is 98%. To me, these additional experiments demonstrate that, as it currently stands, this method is not a convincingly promising framework for "bridging the gap between modern neuroscience and AI". Not to mention, that it appears difficult at the moment to not only implant this into convolutional neural networks (which have been shown in many prior works to yield neurally plausible representations, compared to the fully feedforward networks they try), but also difficulty to scale things up successfully to CIFAR-10, let alone ImageNet.

With these considerations in mind, I cannot recommend this paper for publication.

Response:

We acknowledge and respect the reviewer's concerns regarding the current performance of our Biophysically Detailed Dendritic Neural Networks (BDNN), which yield an accuracy of 52.7% on the CIFAR-10 dataset. It's noteworthy that a conventional ANN, employing an identical architecture and operating under similar conditions, achieved a comparable accuracy of 53.1%.

For added context, we refer to the study by Bartunov et al. (NeurIPS 2018), recommended by the reviewer, that utilized the same architecture on CIFAR-10, achieving an accuracy of 58.7%. However, this performance was accomplished using additional training techniques such as data augmentation, the Adam optimizer, and an extensive training period of 500 epochs. In contrast, our approach, which intentionally avoids these techniques to isolate the contributions of dendrites, relied solely on Stochastic Gradient Descent (SGD) and a significantly shorter training period of 60 epochs.

Furthermore, our adversarial test indicated an improvement of approximately 5% in BDNN's performance compared to a similarly configured ANN. While this might seem modest, it's noteworthy given the simplicity of our approach which intentionally avoids the incorporation of complex training techniques.

We share the reviewer's interest in improving the performance and scalability of BDNNs. With this goal in mind, we are currently refining our multi-GPU framework. This enhancement is expected to pave the way for the implementation of more advanced BDNN architectures and facilitate testing on larger datasets, including ImageNet. As we progress, we remain strong confidence in our commitment to bridging the gap between modern neuroscience and AI through the ongoing development and application of biophysically detailed dendritic neural networks. We envision that BDNNs can offer valuable insights into AI, particularly in elucidating the mechanisms underpinning cognition and computational robustness.

Revisions to the manuscript

As suggested by the reviewers, we have removed the ANN part in the main results and added a brief discussion in the Discussion section,

Line 312-322, "Furthermore, we develop the GPU-based DeepDendrite framework by integrating DHS into CoreNEURON. Finally, as a demonstration of the capacity of DeepDendrite, we present a representative application: examine spine computations in a detailed pyramidal neuron model with 25,000 spines. Further in this section, we elaborate on how we have expanded the DeepDendrite framework to enable efficient training of

biophysically detailed neural networks. To explore the hypothesis that dendrites improve robustness against adversarial attacks, we train our network on typical image classification tasks. We show that DeepDendrite can support both neuroscience simulations and AI-related detailed neural network tasks with unprecedented speed, therefore significantly promoting detailed neuroscience simulations and potentially for future AI explorations.”

Line 437-456, “In response to these challenges, we developed DeepDendrite, a tool that uses the Dendritic Hierarchical Scheduling (DHS) method to significantly reduce computational costs and incorporates an I/O module and a learning module to handle large datasets. With DeepDendrite, we successfully implemented a three-layer hybrid neural network, the Human Pyramidal Cell Network (HPC-Net) (Fig. 6a-b). This network demonstrated efficient training capabilities in image classification tasks, achieving approximately 25 times speedup compared to training on a traditional CPU-based platform (Fig. 6f; Supplementary Table 1).

Additionally, it is widely recognized that the performance of Artificial Neural Networks (ANNs) can be undermined by adversarial attacks—intentionally engineered perturbations devised to mislead ANNs. Intriguingly, an existing hypothesis suggests that dendrites and synapses may innately defend against such attacks. Our experimental results utilizing HPC-Net lend support to this hypothesis, as we observed that networks endowed with detailed dendritic structures demonstrated markedly increased resilience to transfer adversarial attacks compared to standard ANNs, as evident in MNIST and FashionMNIST datasets (Fig. 6d-e). This evidence implies that the inherent biophysical properties of dendrites could be pivotal in augmenting the robustness of ANNs against adversarial interference. Nonetheless, it is essential to conduct further studies to validate these findings using more challenging datasets such as ImageNet.”

We also have modified the content of ANN part in the Abstract, Introduction, and Results:

Line 40-43, “Furthermore, we provide a brief discussion on the potential of DeepDendrite for AI, specifically highlighting its ability to enable the efficient training of biophysically detailed models in typical image classification tasks. “

Line 126-130, “In the discussion we also consider the potential of DeepDendrite in the context of AI, specifically, in creating ANNs with morphologically detailed human pyramidal neurons. Our findings suggest that DeepDendrite has the potential to drastically reduce

the training duration, thus making detailed network models more feasible for data-driven tasks.”

Line 256-258, “Below, we demonstrate how we can utilize DeepDendrite in neuroscience tasks. We also discuss the potential of the DeepDendrite framework for AI-related tasks in the Discussion section.”

REVIEWERS' COMMENTS

Reviewer #3 (Remarks to the Author):

I thank the authors for revising their manuscript and toning down the implications of the results for AI.

I only have one minor suggestion, which is in lines 450-451: "demonstrated markedly increased resilience to transfer adversarial attacks", should be changed to "demonstrated some increased resilience to transfer adversarial attacks". I think "markedly" is too strong of a term given the current results, and especially since these datasets (MNIST variants and CIFAR) aren't overall too challenging.

Once they make this change, their article should be accepted in this revised form, and I wish them success in the pursuit of their DeepDendrite framework!

Dear Reviewers,

Reviewer #3 (Remarks to the Author):

I thank the authors for revising their manuscript and toning down the implications of the results for AI.

I only have one minor suggestion, which is in lines 450-451: "demonstrated markedly increased resilience to transfer adversarial attacks", should be changed to "demonstrated some increased resilience to transfer adversarial attacks". I think "markedly" is too strong of a term given the current results, and especially since these datasets (MNIST variants and CIFAR) aren't overall too challenging.

Once they make this change, their article should be accepted in this revised form, and I wish them success in the pursuit of their DeepDendrite framework!

Response:

We extend our sincere appreciation to the reviewer for highlighting this point, as well as the encouraging words for the future success of the DeepDendrite framework. As advised, we have modified the sentence to reflect a more accurate representation of our findings.

Line 442-444 now reads: "...demonstrated some increased resilience to transfer adversarial attacks compared to standard ANNs, as evident in MNIST and Fashion-MNIST datasets (Fig. 6d-e)."